# Characterization of the COPD alveolar niche using single-cell RNA sequencing

Maor Sauler [1,14✉], John E. McDonough[1,14✉], Taylor S. Adams[1], Neeharika Kothapalli[1], Thomas Barnthaler[1,2], Rhiannon B. Werder[3,4,5], Jonas C. Schupp [1,6], Jessica Nouws[1], Matthew J. Robertson[7], Cristian Coarfa [7], Tao Yang[1,8], Maurizio Chioccioli[1], Norihito Omote[1], Carlos Cosme Jr[1], Sergio Poli [9], Ehab A. Ayaub[10], Sarah G. Chu[10], Klaus H. Jensen [11], Jose L. Gomez [1], Clemente J. Britto [1], Micha Sam B. Raredon [12,13], Laura E. Niklason[12], Andrew A. Wilson[3,4], Pascal N. Timshel[11], Naftali Kaminski [1,14] & Ivan O. Rosas[7,14]

Chronic obstructive pulmonary disease (COPD) is a leading cause of death worldwide, however our understanding of cell specific mechanisms underlying COPD pathobiology remains incomplete. Here, we analyze single-cell RNA sequencing profiles of explanted lung tissue from subjects with advanced COPD or control lungs, and we validate findings using single-cell RNA sequencing of lungs from mice exposed to 10 months of cigarette smoke, RNA sequencing of isolated human alveolar epithelial cells, functional in vitro models, and in situ hybridization and immunostaining of human lung tissue samples. We identify a sub-population of alveolar epithelial type II cells with transcriptional evidence for aberrant cellular metabolism and reduced cellular stress tolerance in COPD. Using transcriptomic network analyses, we predict capillary endothelial cells are inflamed in COPD, particularly through increased CXCL-motif chemokine signaling. Finally, we detect a high-metallothionein expressing macrophage subpopulation enriched in advanced COPD. Collectively, these findings highlight cell-specific mechanisms involved in the pathobiology of advanced COPD.

[1] Pulmonary, Critical Care and Sleep Medicine, Yale School of Medicine, New Haven, CT, USA. [2] Division of Pharmacology, Otto Loewi Research Center, Medical University of Graz, Graz, Austria. [3] Center for Regenerative Medicine of Boston University and Boston Medical Center, Boston, MA 02118, USA. [4] The Pulmonary Center and Department of Medicine, Boston University School of Medicine, Boston, MA 02118, USA. [5] QIMR Berghofer Medical Research Institute, Herston, QLD 4006, Australia. [6] Department of Respiratory Medicine, Hannover Medical School and Biomedical Research in End-stage and Obstructive Lung Disease Hannover, German Lung Research Center (DZL), Hannover, Germany. [7] Pulmonary, Critical Care and Sleep Medicine, Baylor College of Medicine, Houston, TX, USA. [8] Department of Thoracic and Cardiovascular Surgery, The First Affiliated Hospital of Chongqing Medical University, Chongqing, China. [9] Department of Internal Medicine, Mount Sinai Medical Center, Miami, FL, USA. [10] Division of Pulmonary and Critical Care Medicine, Brigham and Women's Hospital, Harvard Medical School, Boston, MA, USA. [11] Intomics A/S, Lyngby, Denmark. [12] Department of Biomedical Engineering, Yale University, New Haven, CT, USA. [13] Medical Scientist Training Program, Yale School of Medicine, New Haven, CT, USA. [14]These authors contributed equally: Maor Sauler, John E. McDonough, Naftali Kaminski and Ivan O. Rosas. ✉email: maor.sauler@yale.edu; john.e.mcdonough@yale.edu

Chronic Obstructive Pulmonary Disease (COPD) is characterized by persistent inflammation and parenchymal lung tissue destruction, particularly in advanced disease stages[1]. COPD is commonly caused by aerosolized pollutants, such as cigarette smoke (CS); however, the clinical and biological effects of CS are heterogenous. Therefore, complex interactions between genetic factors, host responses, and environmental exposures underlie COPD pathogenesis.

Our understanding of COPD pathogenesis has grown over the past decades through the use of murine models, genome-wide association studies (GWAS), and expression analyses using bulk human lung tissue[2-5]. The pathogenic cascade of COPD is initiated by repetitive insults to epithelial and endothelial cells within the distal airways and alveolar niche. This engenders activation of diverse cellular processes including immune cell infiltration, extracellular matrix proteolysis, cellular metabolic dysfunction, loss of proteostasis, DNA damage, autophagy, cellular senescence, and activation of regulated cell death pathways[3,6,7]. Consequently, there is an inability to maintain alveolar homeostasis, chronic inflammation, and alveolar septal destruction, particularly in advanced COPD[3]. However, detailed knowledge of cell-type-specific mechanisms and the complex interactions among multiple lung cell types in COPD is lacking. Recent studies have used single-cell RNA sequencing (scRNAseq) to obtain single-cell resolution and identify disease mechanisms, cellular phenotypes, and changes in alveolar niche crosstalk[8-14]. Therefore, we sought to use scRNAseq to elucidate cell-specific transcriptional profiles of alveolar niche cells in COPD and better understand the pathobiology of this disease.

Herein, we analyze scRNAseq profiles of parenchymal tissue obtained from explanted lungs of patients with advanced COPD requiring lung transplant and control donor lungs. We focus our analysis on three cell types commonly implicated in COPD pathogenesis: epithelial cells, endothelial cells, and alveolar macrophages. Among epithelial cells and based on transcriptomic analyses, we identify a subpopulation of *HHIP*-expressing alveolar epithelial type II (AT2) cells that mediate COPD heritability and have aberrant expression of metabolic, antioxidant, and cellular stress response genes in COPD, including reduced expression of the cellular stress response gene *NUPR1*. Our analyses of endothelial cells suggest capillary CXCL-motif chemokine signaling is an important cause of alveolar inflammation in COPD. Finally, we identify a subpopulation of high metallothionein-expressing alveolar macrophages in the COPD lung.

## Results

We reanalyzed scRNAseq profiles previously obtained from explanted lungs. Lungs were longitudinally sliced in order to sample cells from apical to basal segments and enzymatically digested to obtain single-cell suspension for barcoding and sequencing[12]. Our analysis focused on 17 patients with advanced COPD, and 15 age-matched controls (Fig. 1A). Demographics and pulmonary function test results are shown in Supplemental Table 1. There were eight females in both groups and the median age of all subjects was 62 years old (range 41–80). All COPD subjects had radiographic evidence of advanced emphysema and were former smokers, while four of the donors without COPD were either current or former smokers. Individuals with COPD had a mean ratio of forced expiratory volume in one second (FEV$_1$) to forced vital capacity (FVC) of $0.36 \pm 0.05$ and a mean FEV$_1$ percent predicted of $21.0 \pm 5.0$. The final dataset consisted of 49,976 cells from control lungs and 61,564 cells from COPD lungs with 37 distinct cell types identified in both control and COPD lungs based on representative marker genes (Fig. 1B).

Canonical markers of identified cell types are shown (Fig. 1C–E, Supplemental Table 2). Data were previously deposited in the Gene Expression Omnibus (GSE136831) and can be explored using our online portal (www.copdcellatlas.com).

To validate findings, particularly those in AT2 cells, we performed scRNAseq of isolated lung cells from *Sftpc-CreER^T2^-m^Tm^G* (AT2 cell reporter) mice (Supplemental Fig. 1). Mice were either exposed to 10 months of CS and developed emphysema (Supplemental Fig. 2) or mice were exposed only to room air (RA). The final dataset for this analysis consisted of 19,311 cells from 4 CS-exposed mice (2 male and 2 female) and 20,410 cells from RA-exposed mice (2 male and 2 female). Cellular clusters were manually annotated using previously described cellular markers (Supplemental Fig. 1). We also validated our findings with: (1) RNA sequencing of flow-sorted epithelial cells from single-cell suspensions obtained from 10 subjects with advanced COPD and 16 controls[15]; (2) analysis of microarrays performed on lung tissue parenchyma samples from 208 individuals with COPD generated by the Lung Genomics Research Consortium (LGRC) (GSE47460)[16,17]; (3) in situ hybridization and immunostaining of paraffin-embed tissue samples from control donor lungs or lungs with advanced COPD; and (4) cell culture using induced pluripotent stem cell (iPSC)-derived AT2 cells grown at the air-liquid interface, primary small airway epithelial cells, and A549 cells.

**An AT2 subpopulation contributes to COPD pathobiology and heritability.** Because of their importance in COPD pathobiology, we evaluated the transcriptional profiles of epithelial cell populations in our human scRNAseq dataset. We identified all major epithelial cell types including alveolar type I (AT1), AT2 cells, ciliated cells, goblet cells, and club cells (Fig. 2A). Among these major cell types, there were no significant cell proportion differences between COPD and control lung tissue (Fig. 2B). We also identified 33 recently described aberrant basaloid cells that were discovered in IPF but also occurred to a lesser extent amongst 9 subjects with COPD (1–10 per subject)[12].

We identified two clusters of AT2 cells. These two AT2 clusters had similar transcriptional profiles to the two AT2 clusters reported by Travaglini et al. in their scRNAseq analysis of normal human lungs, and therefore we used similar terms to annotate these clusters[18]. Both AT2 clusters express *SFTPC*, but one cluster was composed of cells with increased expression of canonical "bulk" AT2 markers, such as *SFTPA1*, *SFTPA2*, and *ETV5* (hereon called AT2$_B$ cells) (Fig. 2C, Supplemental Fig. 3). AT2$_B$ cells also had increased expression of *CA2* and increased expression of *WIF1*, a negative regulator of WNT signaling. We compared differentially expressed genes (DEGs) between the two AT2 cell populations and found AT2$_B$ cells had DEGs enriched for gene ontology (GO) pathways related to metabolic processes (FDR $= 4.18 \times 10^{-39}$) and biosynthetic processes (FDR $= 2.64 \times 10^{-37}$). The other AT2 cluster had increased expression of genes related to EGF family receptor signaling (*ERBB4*), WNT signaling (*TNIK*, *TCF12*), and other developmental signaling pathways (*FOXP1*, *STAT3*, *YAP*, and *TEAD1*) (Fig. 2C, Supplemental Fig. 3). Cells from this cluster (hereon called AT2$_S$ cells) had DEGs enriched for GO pathways related to cell differentiation (FDR $= 5.89 \times 10^{-6}$) and cell development (FDR $= 5.92 \times 10^{-6}$). A comparison of the transcriptional profiles of these two AT2 cell populations suggested AT2$_S$ cells may be homologous to AT2 stem and/or alveolar epithelial progenitors (AEPs). However, these similarities are provisional because AEPs are best described in mice and numerous differences between mouse and human AT2 cells have been described[19-21].

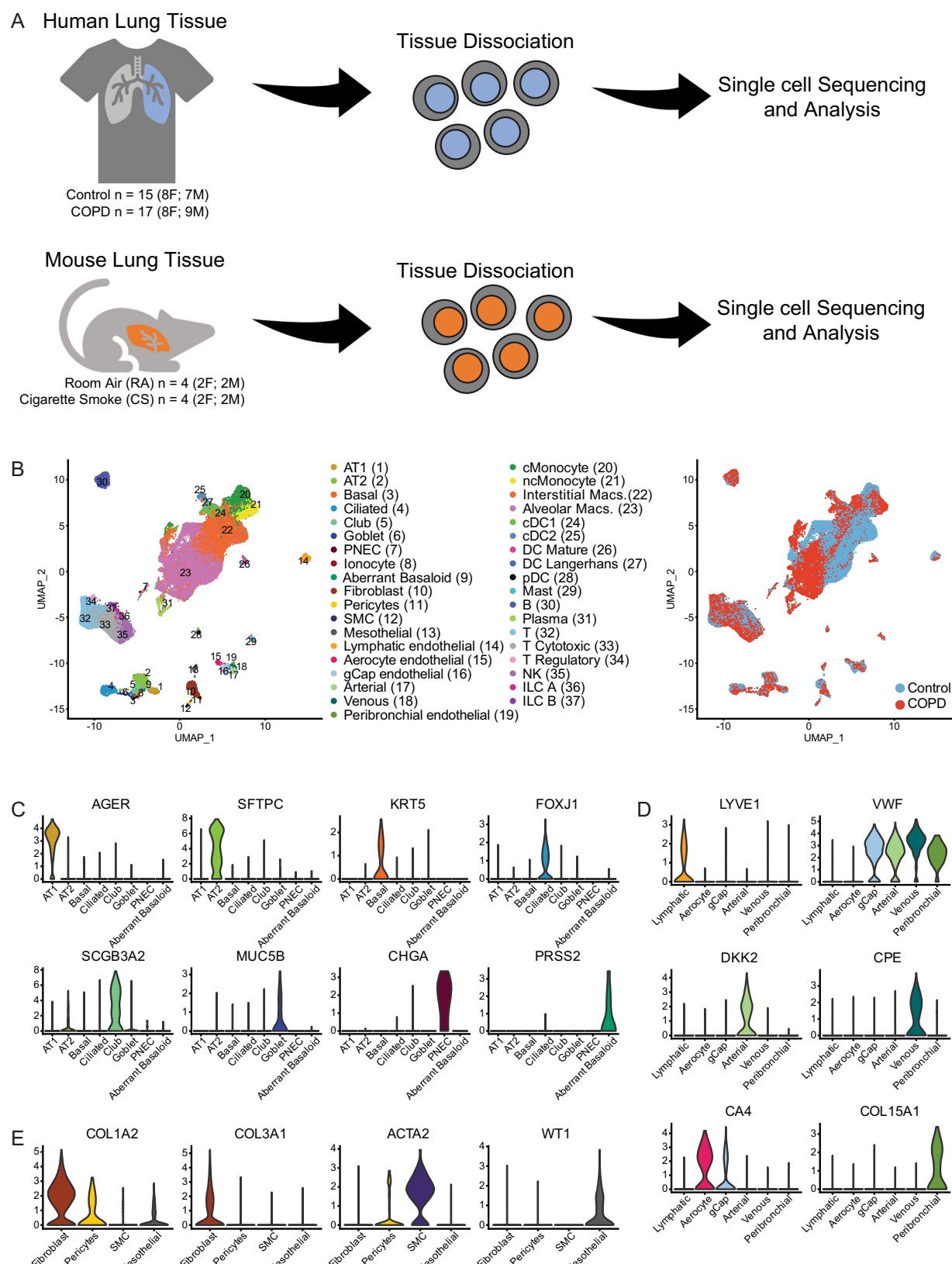

We found $AT2_B$ cells were the predominant expressors of *HHIP* in the lung. *HHIP* is a gene repeatedly identified in COPD GWAS and has been shown to modulate susceptibility to oxidative stress from CS[22–24]. Of the cells expressing *HHIP* (>1 transcript), 80% were $AT2_B$ cells (Supplemental Fig. 4). To validate these findings, we performed in situ hybridization

(Fig. 2D) and quantification of *HHIP* mRNA in SFTPC+ cells. We observed 48% of SFTPC+ cells contained *HHIP* foci and 79% of all *HHIP* foci colocalized with SFTPC+ cells. We then assessed *HHIP* expression in isolated human AT2 epithelial cells obtained by fluorescence-activated cell (FACS) sorting single-cell suspensions of human lung tissue samples (EpCAM^high/PDPN−)[15]. We

**Fig. 1 Profiling of cell types in COPD using scRNAseq data. A** Overview of project design. Tissue from 17 lungs with advanced COPD and 15 control donor lungs were dissociated into single-cell suspensions. Individual cells were barcoded and sequenced for analysis. Similarly, lung tissue from 2 male (M) and 2 female (F) mice exposed to 10 months of cigarette smoke (CS) and 2 M and 2 F mice exposed only to room air (RA) were dissociated into single-cell suspensions, barcoded, and sequenced for analysis. **B** Uniform Manifold Approximation and Projection (UMAP) representation of 111,540 single cells grouped into 37 distinct cell types (left) with identification of COPD and control cells (right). Violin plots of normalized expression values for canonical cell-specific marker genes for **C** epithelial, **D** endothelial, and **E** stromal cells. AT1 alveolar epithelial type I cells, AT2 alveolar epithelial type II cells, PNEC pulmonary neuroendocrine cells, SMC smooth muscle cells, gCap general capillary, cMonocyte classical monocytes, ncMonocyte non-classical monocyte, Macs. macrophages, DC dendritic cells, cDC conventional dendritic cells, pDC plasmacytoid dendritic cells, NK natural killer cells, ILC innate lymphoid cells.

found *HHIP* expression correlated with the expression of AT2$_B$ cell markers including: *SFTPA1*($\rho = 0.94$), *NAPSA* ($\rho = 0.91$), *CA2* ($\rho = 0.91$), *SFTA2* ($\rho = 0.90$), *LAMP3* ($\rho = 0.90$), *SFTPC* ($\rho = 0.90$), *SFTPD* ($\rho = 0.89$), *SFTPA2* ($\rho = 0.88$), *WIF1* ($\rho = 0.88$), *PGC* ($\rho = 0.86$), and *ETV5* ($\rho = 0.85$) (Bonferroni-corrected *P*-value < 0.01) (Fig. 2E). Because *SFTPA1* expression was greater in AT2$_B$ cells and expression of *HHIP* and *SFTPA1* in FACS-isolated AT2 cells were highly correlated, we performed in situ hybridization for *HHIP* and *SFTPA1* mRNA, and immunostaining for SFTPC (Supplemental Fig. 5). We found *HHIP* expression colocalized with *SFTPA1* expression within a subset of SFTPC⁺ cells, but not all SFTPC⁺ cells expressed *HHIP* and *SFTPA1*.

We found AT2$_B$ cells had the largest number of DEGs between control and advanced COPD amongst all epithelial cells (Fig. 2F), underscoring the importance of this cell type in COPD pathogenesis. In addition to *HHIP*, AT2$_B$ cells were major expressors of other genes commonly identified in COPD GWAS including *SERPINA1*[25,26] and *SFTPD*, as demonstrated by a dot plot showing the cell-type-specific expression of top COPD GWAS genes (Supplemental Fig. 6)[27]. A key role for AT2$_B$ cells as mediators of COPD heritability was further supported by the use of CELL-type Expression-specific integration for Complex Traits (CELLECT). CELLECT integrates scRNAseq data with GWAS data and prioritizes cell types enriched for the expression of genes with disease-associated polymorphisms[28]. Using CELLECT, we colocalized our scRNAseq findings with summary statistics from two UKBiobank GWAS evaluating genetic associations with COPD (defined as FEV$_1$/FVC < 70) or lung function (defined as continuous FEV$_1$/FVC)[29,30] (Fig. 2G). AT2$_B$ cells had the greatest enrichment amongst epithelial cells for the expression of genes with polymorphisms associated with COPD-related traits. We also identified high enrichment scores in other epithelial and stromal cell populations, particularly smooth muscle cells, but not immune cells. A summary of the expression-specificity likelihood for each gene across different cell types is shown (Supplemental Dataset 1).

**Altered expression of metabolic and cellular stress response genes in COPD AT2$_B$ cells.** We focused our subsequent epithelial cell analysis on AT2$_B$ cells because of the high COPD-related genetic enrichment and a large number of DEGs. We identified decreased expression of 182 genes and increased expression of 35 genes (Fig. 3A) in AT2$_B$ cells in COPD. Amongst decreased DEGs were multiple genes associated with the electron transport chain including subunits of Complex I (*NDUFAB1*, *NDUFA4*, and *NDUFS8*), Complex III (*UQCRQ* and *UQCRB*), Complex IV (*COX8A*, *COX7B*, *COX6B1*, *COX5B*, *COX7C*, and *COX6C*) and ATP synthase (*ATP5MD*, *ATP5F1D*, and *ATP5F1E*). There were multiple decreased DEGs encoding antioxidants that function in extracellular (*SOD3*), cellular (*GSTO1*, *GSTP1*, and *TXNDC17*), and mitochondrial (*MGST1* and *MGST3*) compartments. We also observed reduced expression of S100A family proteins and genes encoding proteins that mitigate toxin-induced stress (*AKR1A1* and *ALDH2*). Amongst increased DEGs were genes related to

oxidative stress and apoptosis previously implicated in COPD pathogenesis, including *PTPN1*[31] and *EGR1*[32]. Comparison of these DEGs to DEGs between RA- and CS-exposed mouse lung AT2 cells identified two overlapping genes, *NUPR1* and *CD74*. *NUPR1*, a cellular stress response gene and positive regulator of antioxidants[33,34], had the largest reduction in AT2$_B$ expression in COPD (fold change (fc) = 0.35) and was also decreased in AT2 cells from mice exposed to CS (fc = 0.51) (Fig. 3B, C). Expression of *CD74*, a gene encoding a receptor for the cytokine MIF[35], was increased in both human AT2$_B$ (fc = 1.82) and mouse AT2 cells (fc = 1.30) (Supplemental Fig. 7). Collectively, these findings highlight the aberrant expression of key metabolic and stress response genes in the AT2$_B$ cell population in advanced COPD.

We then sought to independently validate decreased *NUPR1* expression in COPD. We identified decreased *NUPR1* expression in FACS-sorted AT2 epithelial cells (EpCAM^high/PDPN⁻)[15] from suspensions of lung cells obtained from COPD and control patients (fc = 0.70) (Fig. 3D). We also found *NUPR1* expression inversely correlated with percent radiographic emphysema (Spearman $\rho = -0.167$) in 208 COPD subjects in the LGRC cohort (Fig. 3E). We then performed immunostaining for NUPR1 and SFTPC and quantified NUPR1 in SFTPC⁺ AT2 cells using CellProfiler, and found AT2 NUPR1 was decreased in COPD lung tissue (Fig. 3F, G, Supplemental Fig. 8).

We next sought to determine the functional significance of reduced NUPR1 expression in COPD AT2 cells. Similar to previous studies demonstrating NUPR1 expression increases in response to oxidative stress, we found expression of NUPR1 increased in response to cigarette smoke extract (CSE) in A549 cells (Supplemental Fig. 9). We then inhibited *NUPR1* mRNA in iPSC-derived AT2 cells grown at air-liquid interface[36], primary small airway epithelial cells, and A549 cells using silencing RNA (Supplemental Fig. 10). Inhibition of *NUPR1* increased susceptibility to CSE-mediated cell death in iPSC-derived AT2 cells (fc = 1.5), primary small airway epithelial cells (fc = 1.9), and A549 cells (fc = 21.3) (Fig. 3H–J, Supplemental Figs. 11, 12). Interestingly, intrinsic apoptosis was not the predominant cell death pathway in *NUPR1*-deficient A549 cells exposed to CSE, as assessed by measurements of caspase-3 and -7 (Supplemental Fig. 13). Therefore, we screened chemical inhibitors of regulated cell death to identify key pathways activated in *NUPR1*-deficient cells exposed to CSE, including a caspase and apoptosis inhibitor (Z-VAD-FMK), a MLKL and necroptosis inhibitor (necrosulfonamide), and an iron chelator and ferroptosis inhibitor (deferoxamine mesylate). We found only deferoxamine reduced susceptibility to cell death in *NUPR1*-deficient A549 cells as assessed by cell cytotoxicity and viability assays (Supplemental Fig. 14) and confirmed this finding using flow cytometry for Annexin V and propidium iodide (Fig. 3I, J). Collectively, these findings demonstrate reduced expression of *NUPR1* in COPD AT2 cells increase their sensitivity to cell death, possibly through increased ferroptosis.

**COPD-specific features of endothelial injury and inflammation.** We identified all major endothelial populations that have been previously described including arterial, venous, lymphatic,

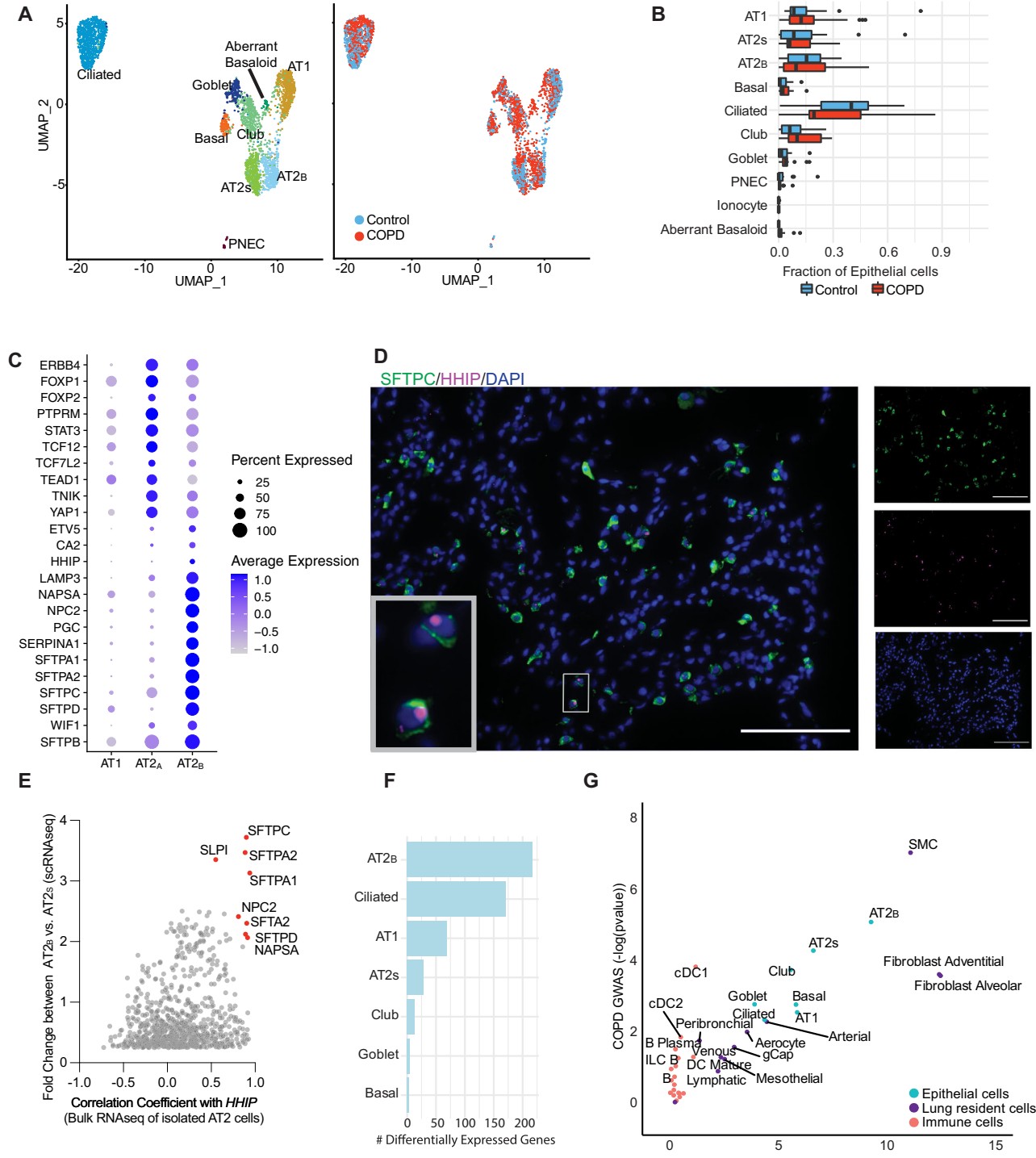

**Fig. 2 COPD epithelial cells and the importance of AT2$_B$ cells in COPD pathogenesis. A** UMAP of epithelial cells from COPD and control donor lungs. Samples are color labelled by cell type (left) and disease category (right). Alveolar type II (AT2) cells can be distinguished as two clusters denoted as AT2$_S$ and AT2$_B$. **B** Distribution of subject-specific epithelial cell types as a fraction of the total number of epithelial cells as assessed by single-cell RNA sequencing (scRNAseq) of 17 lungs with advanced COPD and 15 control donor lungs. Boxes represent median and interquartile ranges (IQRs), whiskers are 1.5 × IQR, and dots represent subjects outside the IQR range. **C** Dot plot of z-scores for marker gene expression values. Dot size reflects percentage of cells with gene expression; color corresponds to the magnitude of gene expression. **D** Immunofluorescence staining for pro-surfactant protein C (SFTPC) (green), in situ hybridization for *HHIP* mRNA(purple), and DAPI (blue) in normal human lung tissue samples. Bar = 100 μm. Original magnification ×20. Results are representative of 5 different samples. **E** Pearson coefficients of genes correlated with *HHIP* expression in isolated AT2 cells (x-axis) and fold change of differentially expressed genes between AT2$_B$ and AT2$_S$ cells in our scRNAseq dataset (y-axis) ($P < 0.05$ using two-sided Wilcoxon rank-sum test with Bonferroni correction). **F** Number of differentially expressed genes between control and COPD across epithelial cell types ($P < 0.05$ using two-sided Wilcoxon rank-sum test with Bonferroni correction). **G** Plot of negative log adjusted P-values for cell-type-specific enrichment for GWAS-identified genes with polymorphisms associated with lung function (continuous FEV$_1$/FVC) (x-axis) and presence of COPD (FEV$_1$/FVC < 70) (y-axis).

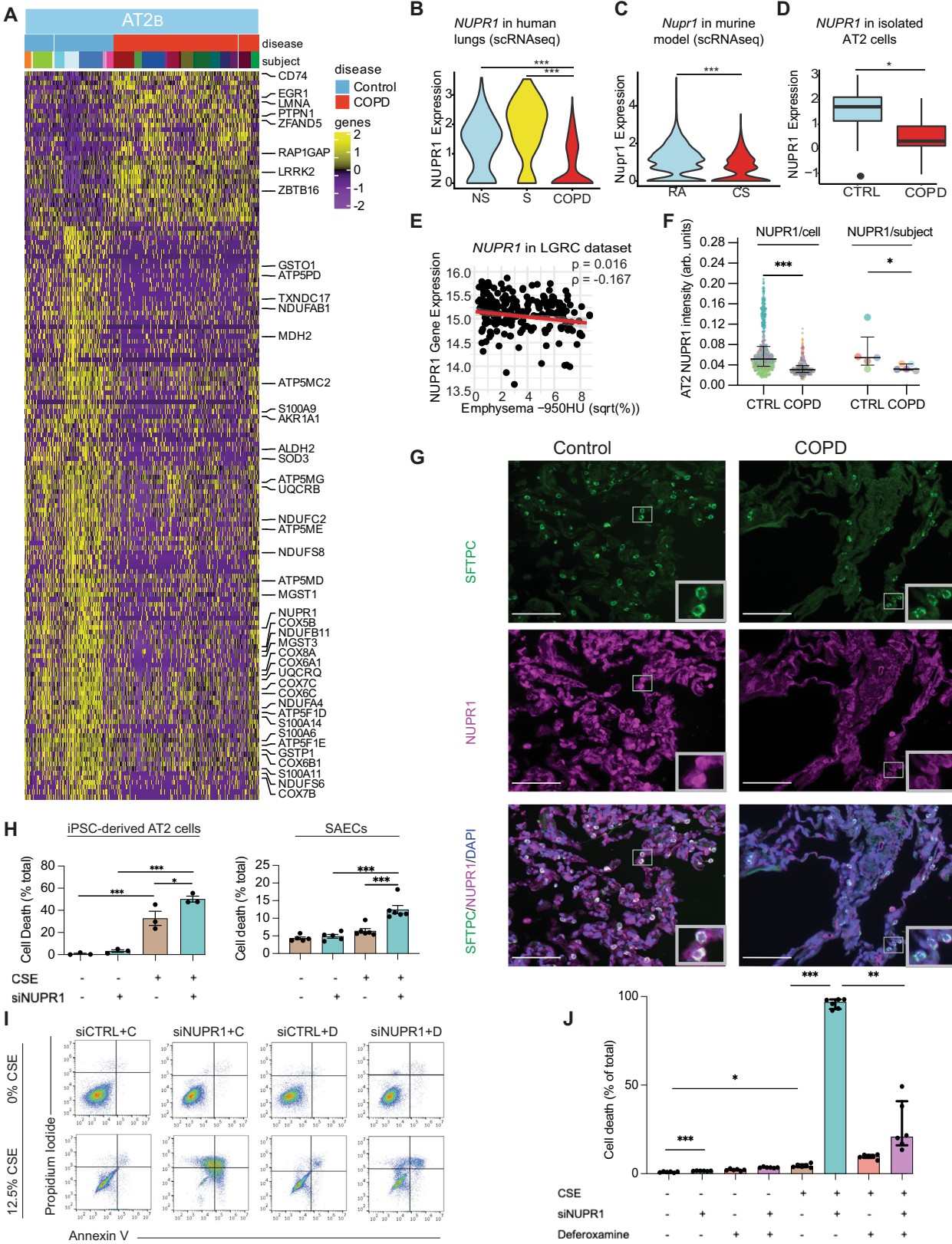

and systemic peri-bronchial endothelial cells, as well as two populations of capillary types. These two capillary cell populations, called aerocytes and general capillaries (gCaps), have the following ascribed functionality based on recently described transcriptional profiling, lineage tracing, and imaging studies[37,38] (Fig. 4A). Both capillary cell types express *PRX* (Supplemental Fig. 15A), but

aerocytes uniquely express *CA4* and *HPGD* and are thought to be specialized for gas exchange and leukocyte trafficking, while endothelial gCaps uniquely express *FCN3* and increased levels of *NOSTRIN* (Supplemental Fig. 15B), and are thought to promote vascular homeostasis by functioning as progenitor cells, modulating vasomotor tone, and regulating immune activation. No

**Fig. 3 Aberrant AT2$_B$ cellular stress response in COPD. A** Heatmap of z-scores for differentially expressed genes between control and COPD for AT2$_B$ cells ($P < 0.05$, two-sided Wilcoxon rank-sum test with Bonferroni correction). Each column represents expression values for an individual cell. Columns are hierarchically ordered by disease phenotype and subject, in which disease category and individual subject are represented by unique colors. z-scores were calculated across all epithelial cells. **B** NUPR1 expression in AT2$_B$ cells from human subjects as assessed by single-cell RNA sequencing (scRNAseq) from former smokers with COPD ($n = 17$), non-smokers without COPD (NS) ($n = 11$), and former/current smokers without COPD (S) ($n = 4$). $P = 2.57 \times 10^{-8}$ (COPD vs. S) and $P = 3.33 \times 10^{-14}$ (COPD vs. NS). **C** Nupr1 expression in AT2 cells from mice exposed to room air (RA) or cigarette smoke (CS) as assessed by scRNAseq ($n = 4$/group). $P = 5.18 \times 10^{-25}$. **D** NUPR1 expression in fluorescence-activated cell sorted (FACS) AT2 cells from control (CTRL) ($n = 16$) and COPD ($n = 10$) subjects. Boxes represent median and interquartile ranges (IQRs), whiskers are $1.5 \times$ IQR, and dots represent subjects. $P = 0.0356$. **E** Two-sided unadjusted Spearman correlation of NUPR1 expression with the square root (sqrt) of radiographic emphysema in lung tissue samples from the LGRC cohort ($n = 208$). **F** Quantification of NUPR1 immunostaining in SFTPC$^+$ AT2 cells in lung tissue from CTRL and COPD lung tissue samples ($n = 5$/group). Each color represents an individual subject. Shown is NUPR1 intensity per AT2 cell ($P < 1.00 \times 10^{-15}$) and mean AT2 NUPR1 intensity per subject ($P = 0.0317$) ($n = 744$ control cells and 662 COPD cells from 5 subjects per group). **G** Sample immunostaining for NUPR1 (purple), SFTPC (green), and DAPI (blue). Bar $= 100 \ \mu m$. Original magnification ×20. Images representative of 5 control and 5 COPD samples. **H** Percent cell death in induced pluripotent stem cell (iPSC)-derived AT2 cells grown at air-liquid interface and small airway epithelial cells (SAECs) treated with NUPR1 silencing RNA (siNUPR1) vs. silencing control RNA (siCTRL) and exposed to 0% or 12.5% cigarette smoke extract (CSE) ($n = 3$/group for iPSC-derived AT2 cells, 5/group for SAECs exposed to 0% CSE, and 6/group for SAECs exposed to 12.5% CSE). **I, J** Flow cytometric detection of propidium iodide (PI) and Annexin V and quantification of cell death (Annexin V$^+$ and/or PI$^+$) in A549 cells exposed to 0% or 12.5% CSE, treated with siNUPR1 vs. siCTRL and deferoxamine mesylate (D) vs. vehicle control (C) ($n = 6$/group). *$P < 0.05$, **$P < 0.005$, ***$P < 0.0001$ using a two-sided Wilcoxon rank-sum test with Bonferroni correction (**B, C**), unadjusted two-sided Wilcoxon rank-sum test (**D, F**), or two-way ANOVA with Tukey post-hoc test (**H, J**). Data are presented as median ± interquartile range (**F, H, J**).

significant differences in the proportion of endothelial cells between disease and control groups were detected (Supplemental Fig. 16). Across all endothelial cell populations, DEGs increased in COPD were enriched for multiple GO pathways including cellular responses to cytokines (FDR $= 2.49 \times 10^{-11}$), cellular responses to stress (FDR $= 2.03 \times 10^{-8}$), and cytokine signaling pathways (FDR $= 2.52 \times 10^{-7}$); while DEGs decreased in COPD were enriched for blood vessel development (FDR $= 4.4 \times 10^{-4}$). We observed many DEGs overlapping across multiple endothelial cell types (Fig. 4B), including increased expression of AP-1 transcription factor subunit genes (*FOS*, *FOSB*, and *JUND*) and decreased expression of genes related to angiogenesis (*ID1*, *ID3*, and *LDB2*) (Fig. 4C). Capillary endothelial cells had the largest number of DEGs involving inflammatory signaling (*CX3CL1* and *IL32*), cellular stress responses (*GADD45B*), and vesicular trafficking (*WASHC2C* and *WASHC2A*); with decreased expression of genes that promote endothelial repair (*SEMA6A* and *WNT2B*). Expression of *TNFRSF10D* and *IRF1* were specifically increased in aerocytes, while expression of *TNFAIP3*, *IFI6*, and *IL6* were specifically increased in gCaps. These findings demonstrate unique and overlapping features of cellular stress and inflammation amongst capillary endothelial cells with advanced COPD.

**Network analyses identified a key role for endothelial CXCL signaling in COPD.** We proceeded to identify capillary endothelial cells as major contributors to alveolar inflammation in COPD using lung connectome analyses[11]. Here, we generated network-level maps of cell–cell signaling across 24 alveolar structural and immune cell types based on computational assessments of predicted ligand–receptor interactions (Fig. 5A). Each node represented a cell population, internodal edges reflected nondirected ligand–receptor interactions, and the edge weight, shown by the thickness of the edge in our network map, represented the sum of all ligand–receptor interactions between nodes identified in the FANTOM5 database[39,40]. We calculated network centrality metrics and edge weights to quantify changes in alveolar signaling topology. Fibroblasts, AT1, and AT2 cells had high degrees of connectivity within the network as reflected by measurements of Kleinberg centrality which prioritize cell types responsible for incoming ("authority") and outgoing ("hub") cell–cell signaling[41]. In contrast, B and T cells had the lowest Kleinberg centrality scores in both control and COPD lung. Further details of the signaling network were observed by

comparing specific ligand–receptor interactions as observed in a comparison of differentially expressed ligands and receptors between AT2$_S$ and AT2$_B$ cells (Supplemental Fig. 17). To further explore changes in cellular communication, we evaluated ligand–receptor interactions that were preassigned to canonical signaling pathways, such as WNT, Sempahorins, and PDGF; thus, allowing us to identify changes in specific cell–cell signaling pathways between control and COPD. We identified changes in outgoing and incoming edge weight among cell types with the top 3 Kleinberg centrality scores in 14 out of 22 pathways (minimum fc > 0.3) (Supplemental Fig. 18, Supplemental Dataset 2). The largest change in edge weight was increased outgoing gCap CXCL-motif signaling (fc $= 234.6$) (Fig. 5B). This finding was specific to COPD and not present in smokers without parenchymal lung disease (Supplemental Fig. 19). Similarly, increased outgoing gCap CXCL signaling was also observed in CS-exposed mice that developed emphysema (Fig. 5C, Supplemental Fig. 20). We then sought to determine the specific ligands and receptors contributing to increased gCap CXCL signaling. A list of CXCL ligand–receptor pairs is shown in Supplemental Table 3. In both mouse and human connectomes, increases in outgoing gCap CXCL signaling were predominantly due to increased expression of *CXCL12* and genes encoding CXCL12-interacting molecules including *CXCR3* and *CXCR4* (Fig. 5C, Supplemental Fig. 21). We validated gCap expression of *CXCL12* in COPD by performing in situ hybridization for *CXCL12* mRNA and immunostaining for PRX and NOSTRIN (Fig. 5D, Supplemental Fig. 22). Collectively, these findings suggest a significant role for outgoing CXCL12 signaling from capillary endothelial cells in COPD.

**A population of alveolar macrophages with increased expression of metallothioneins and *HMOX1* is observed in COPD.** Analysis of immune cell populations revealed significant differences in cell composition among interstitial macrophages (fc $= 0.48$; FDR $= 2.7 \times 10^{-3}$), plasmacytoid dendritic cells (fc $= 3.46$; FDR $= 2.7 \times 10^{-3}$), conventional dendritic cells (fc $= 2.79$; FDR $= 0.025$), and mast cells (fc $= 7.70$; FDR $= 2.7 \times 10^{-3}$) (Supplemental Fig. 23). The largest population of cells were alveolar macrophages which have been implicated in alveolar inflammation and tissue destruction in COPD[42]. To characterize the heterogeneity of alveolar macrophages, we re-embedded the alveolar macrophage cluster in UMAP space and

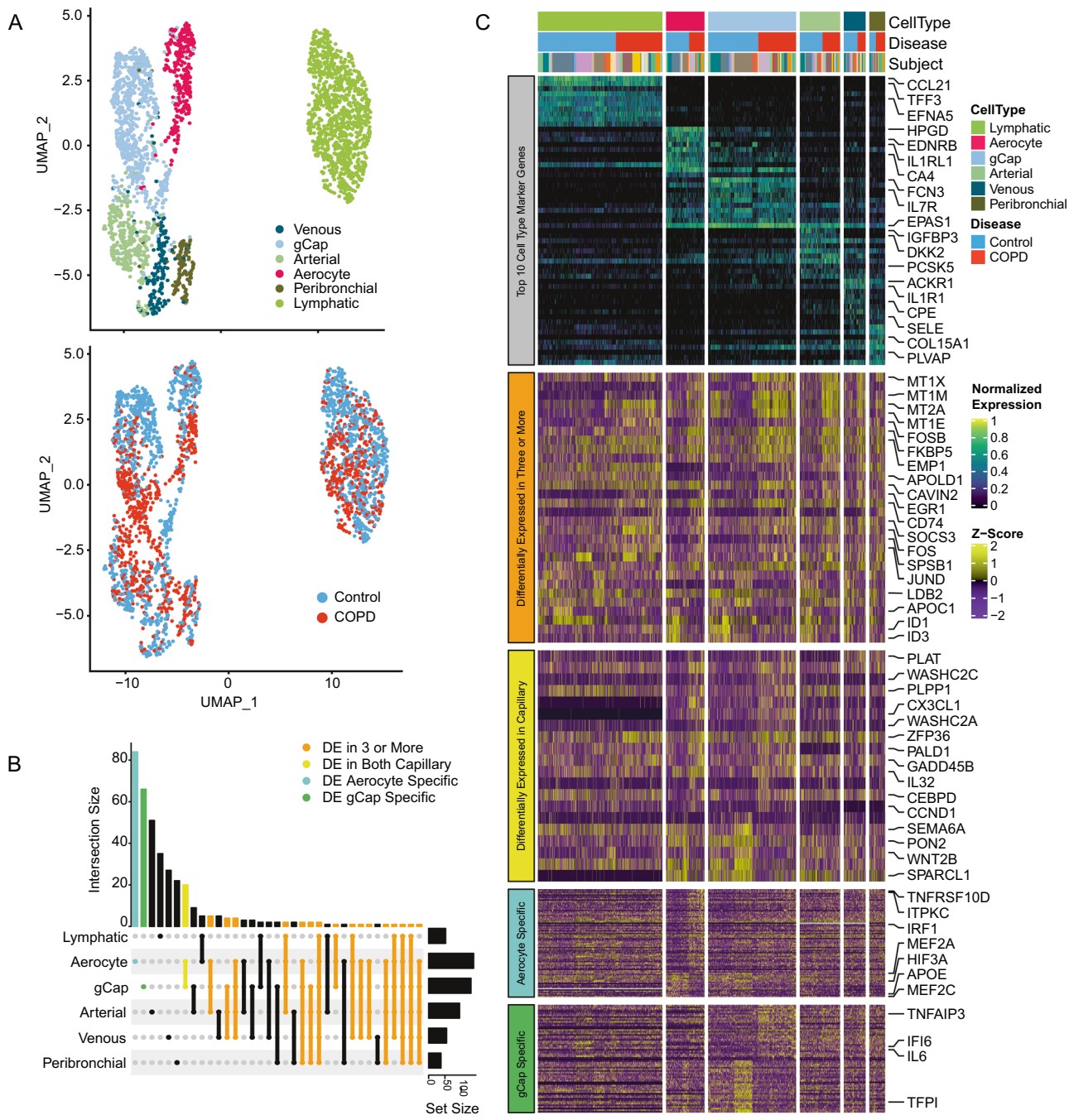

**Fig. 4 COPD endothelial cell types demonstrate universal and cell-type-specific transcriptional aberrations. A** UMAPs of all vascular endothelial (VE) and lymphatic endothelial cells from control and COPD subjects. UMAPs are color labelled by cell type (top) and disease status (bottom). **B** UpSet plot visualizing the properties of intersecting and unique sets of differentially expressed (DE) genes between COPD and control amongst endothelial (two-sided Wilcoxon rank-sum test, unadjusted *P* < 0.001, minimal fold change > 0.5). **C** Heatmap of corresponding differentially expressed genes between COPD and control amongst six subtypes of endothelial cells. Each column represents expression values for an individual cell. Columns are hierarchically ordered by endothelial subtype, disease phenotype, and then subject. Gray row (top): expression values for marker genes are unity normalized between 0 and 1 across all endothelial subtypes. Orange row (middle): z-scores of differentially expressed genes in three or more endothelial cell types between control and COPD. Yellow row (middle): z-scores of differentially expressed genes in both aerocyte and gCap cells between control and COPD. Blue and green row (bottom): z-scores of differentially expressed genes unique to aerocytes (blue) or gCap (green). Unity normalization and z-score calculations were performed using all endothelial subtypes.

reclustered these cells into eight subclusters (Fig. 6A). We then compared the relative abundance of these eight clusters between control and COPD and identified two changes in alveolar macrophage population composition between COPD and control; cluster-0 macrophages were enriched amongst controls while cluster-5 macrophages were enriched among patients with

COPD (FDR < 0.05) (Fig. 6B, Supplemental Fig. 24). The corresponding cluster markers for cluster-5 and cluster-0 are shown in Fig. 6C. The top cluster-5 markers were metallothioneins (*MT1G, MT1X, MT1E, MT2A, MT1M, MT1F, MT1H, MT1A,* and *MT1L*), with macrophages expressing elevated levels of these cysteine-rich metal-binding antioxidant proteins, and

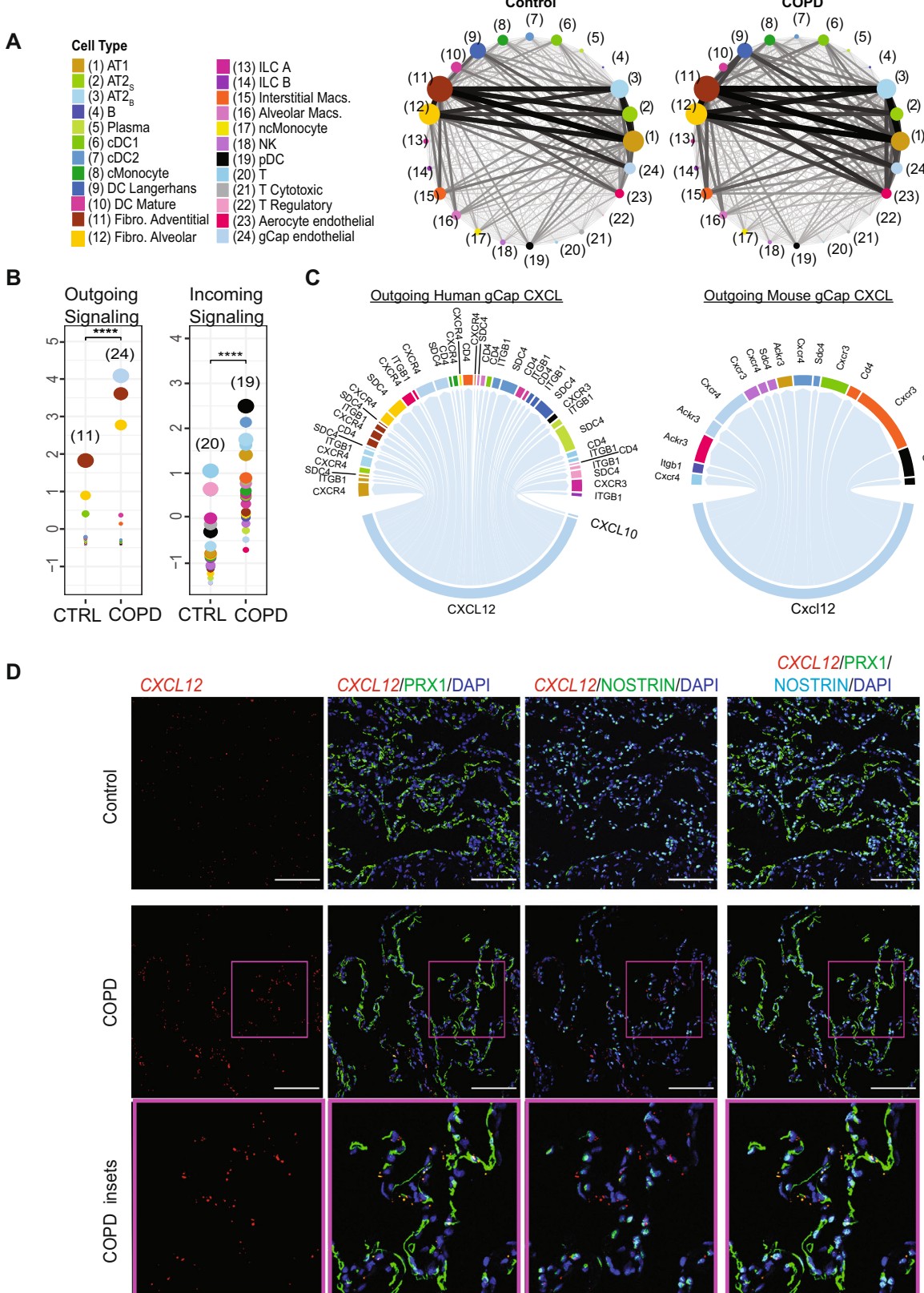

increased expression of *HMOX1*, a target gene of the antioxidant transcription factor NRF2[43,44]. We confirmed increased MT2A in a subset of cells co-expressing the phagocyte cell marker CD68 in COPD lung tissue samples (Fig. 6D–F, Supplemental Fig. 25). There were also multiple DEGs between COPD and control macrophages that were not represented in any specific cluster. The most significant DEGs were associated with chemotaxis and inflammation. These include *THBS* (fc = 1.66), *PELI1* (fc = 1.55), and *CDC42* (fc = 1.34) (Fig. 6G, Supplemental Fig. 26). These results highlight changes in macrophage population composition, including the presence of metallothionein-

**Fig. 5 Alveolar niche networks and pathway centrality analyses. A** Network plots of the alveolar niche in control (left) and COPD (right). Each node represents a cell population and each internodal edge reflects ligand–receptor interactions between cell types. The edge weight (thickness) between nodes reflects the sum of individual edge weights (nondirected) which are based on the product of ligand–receptor gene expression values, while the size of the node reflects measurements of Kleinberg centrality which prioritizes cell types responsible for incoming (authority) and outgoing (hub) cell–cell signaling. Individual cell types are labelled by color and number. **B** Centrality analysis of the alveolar connectome for CXCL signaling between control and COPD. Dot size is proportional to the Kleinberg scores for each cell type within CXCL signaling. Panel shows outgoing edge weights and Kleinberg hub scores (left) and incoming edge weights and Kleinberg authority scores (right). Individual cell types are color labelled as in panel **A**, and numbers shown identify cell types with the largest Kleinberg centrality score. ***$P < 0.0001$ using a two-sided Durbin test to compare control and COPD across cell types. **C** Differential circos plots for outgoing gCap CXCL signaling from human and mouse connectomes. Edge thickness is proportional to perturbation scores, defined as the product of the absolute values of the log-fold change for both the receptor and ligand. CXCL differential network analysis limited to edges in which both ligand and receptor expression are increased. **D** Immunofluorescence staining for PRX (green), NOSTRIN (green or aqua), in situ hybridization for *CXCL12* mRNA (red), and DAPI (blue) in normal and COPD human lung tissue samples. Bar = 100 μm. Original magnification ×20. Images representative of 5 control and 5 COPD samples.

expressing alveolar macrophage subpopulation in advanced COPD, as well as generalized DEGs across multiple macrophage subpopulations.

## Discussion

COPD pathogenesis involves diverse and heterogenous biologic processes that vary across cell types but culminate in the loss of alveolar homeostasis and chronic inflammation. The extent to which pathologic COPD-related mechanisms manifest in specific cell types remains uncertain. In this study, we analyzed single-cell transcriptional profiles of lung alveolar cells and identified transcriptional cell-specific profiles that may contribute to the pathobiology of advanced emphysema in COPD patients. We showed transcriptional evidence for altered bioenergetics and impaired cellular stress tolerance in AT2$_B$ cells, including decreased *NUPR1* expression. We found endothelial cells had increased inflammatory expression profiles, including aberrant gCap CXCL12 signaling. We also identified a high metallothionein-expressing macrophage subpopulation enriched in advanced COPD. These findings provide single-cell resolution of COPD pathobiology within the alveolar niche.

Our findings highlight an important role for AT2$_B$ cells in COPD pathobiology. AT2$_B$ and AT2$_S$ cells are subtypes of AT2 cells that we identified in our scRNAseq analysis of human lung tissue samples and were recently described by Travaglini et al.[18]. AT2$_S$ cells had increased expression of genes related to cellular development and therefore may represent AEPs, while AT2$_B$ cells had increased expression of canonical AT2 genes. Interestingly, the largest difference between the two cell populations was increased AT2$_B$ expression of genes implicated in surfactant production and other homeostatic AT2 functions, suggesting that contrasting features between the two AT2 cell populations are not only related to AT2 progenitor function, and therefore their different properties may extend beyond the eponymous functions of AEPs and non-AEPs. Additionally, it should be noted that while AEPs are best described in mice, we did not identify two clear AT2 clusters in mice as we identified in humans. There are also substantial differences between mouse and human AT2 cells[19–21]. Therefore, future mechanistic studies will be necessary to elucidate the function of AT2$_S$ and AT2$_B$ cells in the human lung.

We found AT2$_B$ cells had the largest number of DEGs amongst epithelial cells in advanced COPD. These genes were related to mitochondrial metabolism and redox regulation, including decreased expression of genes encoding antioxidants, electron transport chain complexes I, III, and IV, and ATP synthetase. Previous studies have demonstrated alveolar epithelial mitochondria are dysmorphic in COPD, and identified changes in alveolar mitochondrial biogenesis, mitophagy, fission, and fusion with COPD[45–47]. Such changes can impair ATP production and

increase mitochondrial reactive oxygen species generation, which in the setting of reduced antioxidants, can promote hallmark COPD pathobiological findings including oxidative stress, cellular injury, and tissue destruction[6]. While aberrant bioenergetics, impaired redox regulation, and increased cellular injury are well-associated with COPD pathogenesis, our analyses localized these findings to AT2$_B$ cells.

Our integrative analysis of COPD GWAS with our scRNAseq data (CELLECT) also identified AT2$_B$ cells as key epithelial mediators of COPD heritability, in part because AT2$_B$ cells are major expressors of *SERPINA1* (along with monocyte and macrophage populations), *SFTPD*, and *HHIP*. Previous GWAS firmly established an association between *HHIP* polymorphisms and COPD-related traits[23]; while animal studies have shown haploinsufficient *Hhip* mice are prone to oxidative stress and emphysema[24]. Subsequent studies using these haploinsufficient *Hhip* mice demonstrated reduced antioxidant capacity and alterations in cellular metabolism[48]. Importantly, our data show *HHIP* is predominantly expressed by AT2$_B$ cells in human lungs. While we did not identify any differences in AT2$_B$ expression of *HHIP* between COPD and control, alterations in cellular metabolism and antioxidant capacity described in *Hhip* haploinsufficient mice may be reflected by parallel findings in COPD AT2$_B$ cells. However, it should be noted that in contrast to adult humans, adult mice predominantly express *HHIP* in a subset of fibroblasts rather than AT2 cells[49]. Therefore, the small degree of *HHIP* expression in other adult human lung cell types or during development may also influence COPD susceptibility in a manner unrelated to AT2 cells[50,51]. The potential mechanistic links between *HHIP* polymorphisms, altered AT2$_B$ bioenergetics, impaired cellular stress responses, and susceptibility to COPD will require further investigation. Another observation related to our CELLECT analysis was stromal cells and epithelial cells, but not immune cells, were enriched for genetic variants associated with decreased lung function. This raises the intriguing possibility that aberrant immune cell phenotypes are acquired rather than genetically programmed in COPD patients. Therefore, despite the inflammatory nature of COPD, targeting epithelial (particularly AT2$_B$ cells) and stromal cells in at-risk individuals with early COPD may be an alternative therapeutic strategy.

We also identified decreased AT2$_B$ expression of *NUPR1*, a finding that we validated using human isolated AT2 cells, quantitative immunostaining, independent datasets, and comparative murine studies. NUPR1 is a cellular stress response transcription factor that promotes chemoresistance in multiple cancer cell lines and protects against toxin-induced renal epithelial injury[34,52]. While the role of NUPR1 in non-malignant lung disease remains uncertain, an association between *NUPR1* and COPD was recently suggested by Morrow et al. who integrated findings from whole-genome methylation profiling of

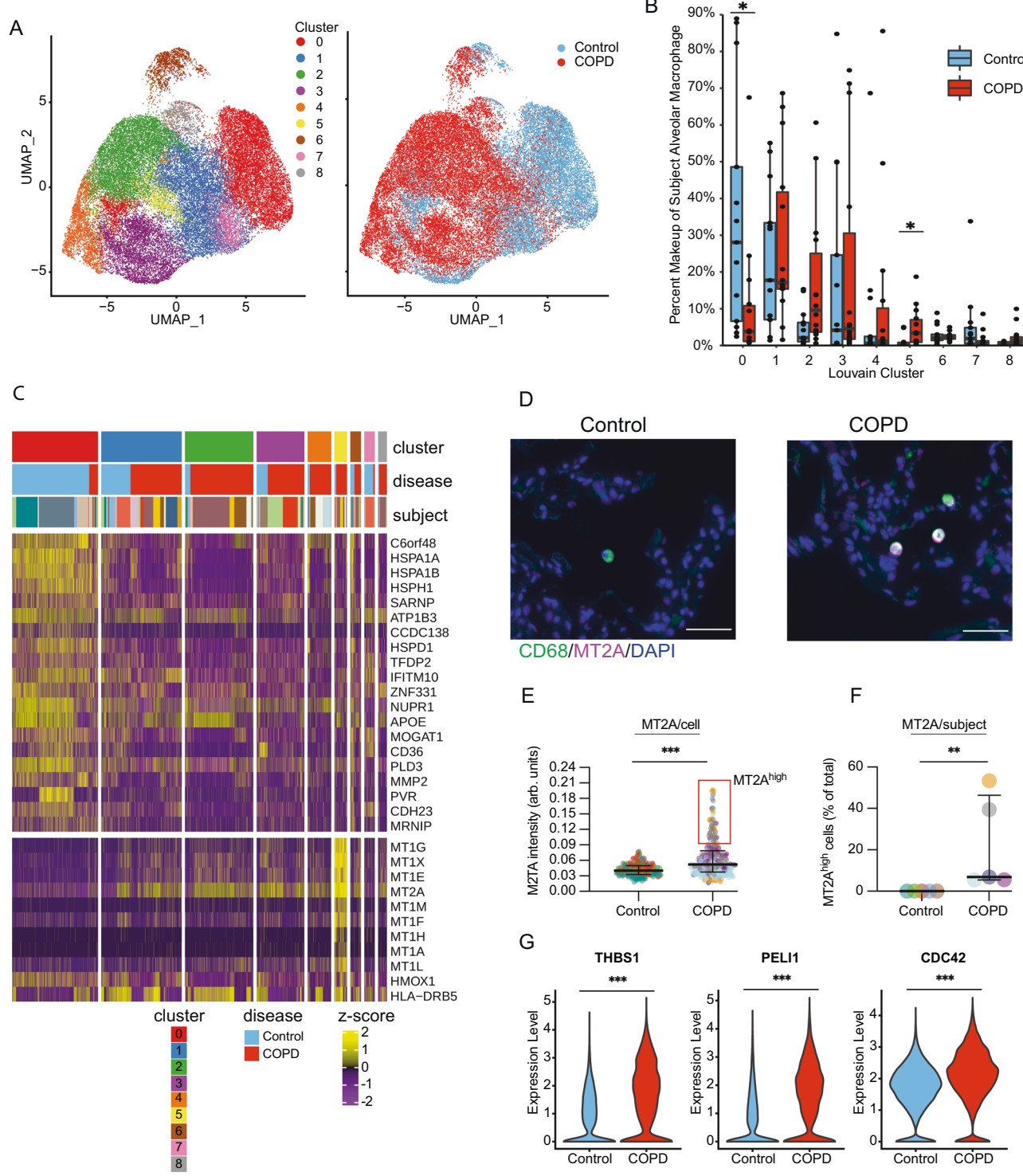

human lung tissue with COPD GWAS[53]. They colocalized single-nucleotide polymorphisms (SNPs) identified in COPD GWAS with local *cis*-regulation of CpG methylation and suggested that SNPs in close proximity to *NUPR1* mediate local epigenetic regulation with relevance to COPD. Additionally, analyses by Hedstrom et al. suggest extracellular matrix in COPD may exert diverse effects including decreased *NUPR1* expression[54]. Using three different in vitro models, including iPSC-derived AT2 cells grown at the air-liquid interface, we showed inhibition of *NUPR1* mRNA increased susceptibility to CSE-mediated cell death.

Recent studies demonstrated NUPR1 suppresses ferroptosis, an iron-dependent regulated cell death pathway. In A549 cells, the increased susceptibility to cell death of *NUPR1*-deficient cells was reversed by using deferoxamine mesylate, an iron chelator and ferroptosis inhibitor. Aberrant iron metabolism and ferroptosis have been recently implicated in COPD pathogenesis[47,55]. Ferroptosis is also associated with decreased antioxidant capacity and the accumulation of lipid reactive oxygen species (ROS)[56]. Therefore, decreased $AT2_B$ expression of *NUPR1* coupled with aberrant $AT2_B$ bioenergetics may increase susceptibility to

**Fig. 6 Changes in alveolar macrophage population composition in COPD. A** UMAPs of control and COPD alveolar macrophage cells color labelled by Louvain cluster (left) and disease status (right). **B** Percent makeup of alveolar macrophage across all nine Louvain clusters per subject, as assessed by single-cell RNA sequencing (scRNAseq) from 14 lungs with advanced COPD and 13 control donor lungs. Boxes represent median and interquartile ranges (IQRs), whiskers are 1.5 × IQR, and dots represent subjects. $P = 0.0281$(cluster-0) and 0.0231 (cluster-5). **C** Heatmap of the distribution of z-scores of marker genes for cluster-0 and cluster-5 alveolar macrophages (*$P < 0.05$, two-sided Wilcoxon rank-sum test with Bonferroni correction). Columns represent expression values from individual cells and are hierarchically ordered by macrophages cluster, disease status, and subject. **D** Sample immunofluorescence images of MT2A (purple) in CD68[+] cells (green) in control and COPD lung tissue samples (arrows). Arrows point to colocalization of MT2A and CD68 (white). Scale bar = 50 μm. Original magnification of cropped image = ×20. Images representative of 5 control and 5 COPD samples. **E** Quantification of MT2A immunofluorescence staining in CD68[+] cells ($n = 233$ control cells and 277 COPD cells from 5 subjects/group). Each color represents a different patient. Data are presented as median ± interquartile range. $P = 1.39 \times 10^{-13}$. **F** Quantification of the percent of MT2A$^{high}$ cells per patient ($n = 5$/group). Data are presented as median ± interquartile range. $P = 0.00794$. **G** Violin plots of *THBS1*, *PELI1*, and *CDC42* expression in alveolar macrophages in control and COPD subjects. $P = 1.57 \times 10^{-31}$ (*THBS1*), $2.63 \times 10^{-28}$ (*PELI1*), and $5.77 \times 10^{-13}$ (*CDC42*). *$P < 0.05$, **$P < 0.005$, ***$P < 0.0001$ using Wilcoxon rank-sum test adjusted for FDR (**B**), unadjusted two-sided Wilcoxon rank-sum test (**E**, **F**), and two-sided Wilcoxon rank-sum with Bonferroni correction (**G**).

ferroptosis in these cells. Future studies will be necessary to dissect the complex interplay between oxidative stress, NUPR1, and iron metabolism in COPD.

We found a role for endothelial cells as key contributors to persistent alveolar inflammation in advanced COPD. Endothelial injury is a well-established pathogenic mechanism of advanced COPD[57], and emphysema severity is associated with serologic and radiographic evidence of pulmonary vascular dysfunction and disease[57,58]. In mice, emphysema can be induced by inhibiting VEGF signaling, which is critical for endothelial homeostasis[59], and multiple studies have shown the endothelial injury is necessary and sufficient to cause emphysema[60]. Our transcriptome analysis provides further evidence for reduced endothelial homeostasis in advanced COPD as evidenced by reduced expression of endothelial maintenance factors and increased endothelial expression of cellular stress response genes and AP-1 transcription factors. However, the most intriguing finding was the increase in capillary CXCL12 signaling. While most studies have focused on the interaction between epithelial and immune cells as the critical cause for persistent inflammation despite smoking cessation[61], we identified increased outgoing gCap CXCL-motif chemokine signaling as the largest change in cell–cell communication in the lung with advanced COPD. Increased CXCL chemokine signaling has been previously implicated in lymphoid neogenesis and COPD pathogenesis[62], and a recent study demonstrated that AMD3100, a drug that inhibits the CXCL12 receptor CXCR4, reduces CS-induced emphysema in a mouse model[63]. Prior single-cell analyses have shown pulmonary arteries, but not pulmonary capillary, express *CXCL12* in healthy human lung tissue[64]. However, in COPD our study prioritized gCap cells as a major source of outgoing endothelial CXCL12 signaling, a finding we also demonstrated in mice and validated through in situ hybridizations of *CXCL12* mRNA in COPD lung tissue. We identified multiple potential target cell populations for CXCL signaling but future mechanistic studies are required to determine the molecular attributes and therapeutic potential of gCap CXCL12 inhibition.

In contrast to solely identifying phenotypic shifts amongst endothelial cells with advanced COPD, we observed both phenotypic shifts and population composition changes amongst alveolar macrophages. A subset of macrophages in control lungs was enriched for heat shock proteins while lungs from subjects with advanced COPD were enriched for a cluster with increased expression of metallothionein. While previous scRNAseq analyses of alveolar macrophages have identified alveolar macrophages that express metallothionein[65], we found an alveolar macrophage subset where the magnitude of metallotheinein expression was elevated in COPD. Previous studies have shown metallothioneins are induced by oxidative stress and inflammation, and protect

against cellular injury by sequestering intracellular metals such as zinc and copper[66]. Metallothioneins have protective roles in acute lung injury models[67], but little is known about the consequences of chronic metallothionein upregulation in the setting of advanced COPD where redox homeostasis and heavy metal metabolism are disrupted[68]. However, aberrant metabolism of intracellular metals is already implicated in COPD pathogenesis. For instance, Menkes disease, a congenital disorder of copper deficiency, causes emphysema[69]. Dysregulated zinc homeostasis can cause impaired phagocytosis and abnormal inflammatory response in macrophages[70]. Therefore, understanding the metallothionein regulation of zinc and copper in COPD may improve our understanding of disease pathogenesis.

There are a few limitations that must be considered while contemplating the findings of this study. First, transcriptional changes are informative but do not always reflect protein concentration or function. While we validated key findings, further studies will be needed to fully elucidate the mechanisms through which the identified transcriptional phenotypes of alveolar cells in COPD contribute to disease pathogenesis. Second, COPD is a heterogenous disorder, yet the subjects in this study represented a distinct subtype of COPD patients with advanced emphysema requiring transplant and who were not actively smoking. Therefore, our findings may not be generalizable to all COPD subphenotypes or earlier disease stages. While all COPD subjects had radiographic emphysema, we do not know the degree of emphysema at the specific location from where the tissue was sampled and cannot be sure that unrecognized systemic biases in sample processing did not influence the results of our findings. However, samples were processed in a stereotyped fashion in which lung samples were longitudinally sliced from apical to basal segments to minimize any sample processing biases and biases caused by spatial heterogeneity. Additionally, the full spectrum of pathologic parenchymal changes was likely captured in our analysis given the large number of subjects in our study. There are also limitations common to scRNAseq experiments that deserve mention. These include "dropout" in which transcripts, particularly from lowly expressed genes, are not detected, causing the data to be sparse and zero-inflated. Consequently, our analyses are more likely to be biased towards detecting differences among highly variable genes that are less affected by dropout. Another important limitation is dissociation bias due to variable cellular sensitivities to digestive enzymes and differences in how cells are embedded in the extracellular matrix. Such biases can lead to cellular proportions that are different from those found in vivo. As an example, dissociation biases contributed to the sequencing of only 631 AT1 cells (average of 20 cells/subject) but 22,998 interstitial macrophages (average of 719 cells/subject), which we know isn't representative of what is in the lung. However, all

samples were processed using previously validated protocols[15], hundreds of cells from each cell type were sequenced, and there was no singular subject that contributed overwhelmingly to specific cell types that would bias the results. Limitations in our connectome analyses include a lack of assessments related to the spatial relationships amongst inferred cell–cell networks, and therefore, findings need to be interpreted cautiously. However, we were able to detect increased CXLC12 gCap signaling in both mice exposed to CS and humans with COPD, which we validated using in situ hybridization.

Collectively, our findings provide a high-resolution single-cell atlas of the alveolar niche in the COPD lung. This atlas identified previously unrecognized changes in gene expression and cellular interactions in distinct epithelial, endothelial, and macrophage cell populations in COPD, highlighting the complexity and diversity of cellular injury and inflammation in COPD. Future studies evaluating outcomes of specific transcriptional dysregulations identified herein will provide insights into mechanisms that contribute to disease.

## Methods

**Ethical approval**. Study protocols for research related to human samples including informed consent, publication of demographic data, and associated corresponding source data were approved by Partners Healthcare Institutional Board Review (IRB Protocol 2011P002419). Animal protocols were approved by the Animal Care and Use Committee at Yale University (2019-07867).

**Human tissue sample scRNAseq**. We reanalyzed scRNAseq of parenchymal lung tissue previously published by Adams et al., focusing on individuals with and without COPD and excluding samples from individuals with pulmonary fibrosis[12]. One COPD sample was excluded due to a reported history of no cigarette smoke exposure. In order to ensure our control and COPD samples were age-matched, we excluded control samples from individuals <40 years of age. Tissue procurement, sample processing, and single-cell sequencing methods for these samples have been previously described[12,15]. Briefly, explanted lungs were procured from donors with end-stage lung disease undergoing transplant or rejected control donor lungs. Explanted organs were longitudinally sliced, and three to four longitudinal biopsies, containing tissue from apical to basal segments of the lung, were washed with cold sterile phosphate-buffered saline (PBS) and visible airway structures, vessels, blood clots, and mucin were removed. Lung tissue was mechanically minced, enzymatically digested, and cryopreserved. Single-cell barcoding of thawed samples and complementary DNA (cDNA) library preparation was performed according to the manufacturer's protocol (Single Cell 3′ Reagent Kits v2, 10x Genomics, USA). Quality control was maintained using an Agilent Bioanalyzer High sensitivity DNA chip. The cDNA libraries were sequenced on a HiSeq 4000 Illumina platform, with a goal of 150 million reads per library and a sequencing configuration of 26 base pairs on Read1 and 98 base pairs on Read2. Full de-identified sequencing data for all subjects are available in the gene expression omnibus (GEO) under accession number GSE136831. Base call files were demultiplexed into FASTQ files using Cell Ranger's (v3.0.2) mkfastq pipeline. The Read2 files were trimmed using cutadapt (v2.7) and read shorter than 20 bp were removed. Read processing was conducted with zUMIs v2.0 pipeline and trimmed reads were aligned to the GRCh38 release 91 (GRCh38.p12) using STAR (v2.6.0c). We removed barcoded cells with <12% of transcripts arriving from unspliced mRNA, cells with <1000 transcripts profiled, and cells with >20% of their transcriptome of mitochondrial origin. Expression values were normalized to 10,000 transcripts per cell and log-transformed using a pseudocount of 1.

**Mouse scRNAseq**. From Jackson Laboratories, we obtained *Sftpc-CreER^T2* (stock #028054) and *Rosa26-m^TmG* C57Bl/6 (stock #007676) mice, and bred them together to generate *Sftpc-CreER^T2-m^TmG* mice. *Sftpc-CreER^T2* mice were previously described by Rock et al.[71] and *Rosa26-m^TmG* mice were previously described by Muzumdar et al.[72]. Male and female 8–10-week-old mice received tamoxifen (T5648; Sigma–Aldrich) (20 mg/mL stock solution in corn oil) at 150 mg/kg × 4 days given via intraperitoneal injection. Two weeks later, littermates were randomly assigned to begin exposure to room air (n = 2 males and 2 females) or cigarette smoke from 3R4F research cigarettes (University of Kentucky) in a Teague TE-10 smoking machine (Teague Enterprises) (n = 2 males and 2 females) for 6 h of exposure per day, 5 days/week for 10 months. One day after the last exposure, single-cell suspensions were obtained by placing right lungs from PBS perfused mice in digestion media (DMEM containing 1 mg/mL Collagenase/Dispase (Roche), 3 U/mL of Elastase (Worthington), and 20 U/mL of DNAase (Qiagen) and incubating at 37 °C for 45 min. Digested lung tissue was meshed through a 100 μm cell strainers using a plunger and resuspended in 20 mL ice-cold DMEM + 10% FBS. ACK lysis buffer was used to remove red blood cells. To enrich non-immune cells, single-cell suspensions were MACS sorted on an LS column following incubation with CD45+ microbeads (Milltenyi) for 15 min at 4 °C per protocol. Samples were filtered through 40 μm filters, and viable cells were counted using a Countess II automated cell counter (Thermo Fisher). Cell populations were reconstituted to achieve a final concentration of 10^6 cells/mL consisting of 10% CD45+ and 90% CD45- populations. Single-cell barcoding and complementary DNA (cDNA) library preparation were performed according to the manufacturer's protocol (Single Cell 3′ Reagent Kits v3, 10x Genomics, USA). Briefly, cell suspensions, beads, master mix, and portioning oil were loaded on to single-cell "A" chip for a targeted output of 10,000 cells per library and run on the Chromium Controller. Reverse transcription was performed at 53 °C for 45 min and cDNA was amplified for 12 cycles using a BioRad C1000 Touch thermocycler. We performed cDNA size selection using SpriSelect beads (Beckman Coulter, USA) and cDNA quality was confirmed with an Agilent Bioanalyzer High Sensitivity DNA chip. DNA fragmentation, end-repair, A-tailing, and ligation of sequencing adapters were performed per the manufacturer's protocol (10x Genomics, USA). The cDNA libraries were sequenced on a HiSeq 4000 Illumina platform aiming for 150 million reads per library and a sequencing configuration of 28 base pairs on Read1 and 98 base pairs on Read2. Base call files were demultiplexed into FASTQ files using the mkfastq pipeline in Cell Ranger (v3.0.2). Adaptor contamination (AAGCAGTGGTATCAACGCAGAGTACATGGG 10×3-prime samples and 20 bp or longer poly(A) sequences were removed using cutadapt (v2.9). and reads shorter than 25 bp were removed. Read processing was performed using STAR (v2.7.3a). and aligned to the mouse reference genome GRCm38 release M22 (GRCm38.p6) downloaded from GENCODE[73]. A modified genome index for the STAR alignment was created by adding the nucleotide sequence information of the transgenes *eGFP* and *tdTomato*, to which sequencing reads were then aligned. Collapsed unique molecular identifiers (UMIs) with reads originating from spliced as well as unspliced RNA were retained. We removed barcoded cells with <7.5% of transcripts arriving from unspliced mRNA, cells with <1000 transcripts profiled, and cells with >5% of their transcriptome of mitochondrial origin. Background contamination from cell-free mRNA was removed using SoupX software (v1.2.2)[74].

**Clustering, differential cell expression, and cell population composition**. Clustering and differential analysis of cells was performed using the Seurat package (v.3.2.0 and v. 4.0.2.) in R[12]. Expression values were normalized to 10,000 transcripts per cell and log-transformed using a pseudocount of 1. Regression during scaling was performed to adjust for percent mitochondrial genes. Louvain clustering was used to group cells and cell type clusters were identified using canonical marker genes. Overall marker genes for each cell type were identified by applying the Seurat FindAllMarkers implementation of the Wilcoxon rank-sum test or by calculating the diagnostics odds ratio (DOR) for each gene per cell type as previously described[12]. Cells annotated as "Multiplet" were removed prior to downstream analyses. Differentially expressed genes between COPD and control cells were identified by using the FindMarkers function test in Seurat, with statistical test and adjustment for multiple comparison testing as described in the methods and figure legend. Gene Ontology analyses were performed with AmiGO[75,76] (Fisher Exact Test with FDR correction). For endothelial cells, UpsetR plots were generated using the UpsetR package[77].

**CELL-type expression-specific integration for Complex Traits (CELLECT)**. We used CELLECT (v.1.1.0) to quantify associations between cell-type specificity of expressed genes and findings from genome-wide association studies (GWAS) of lung function and COPD[29,30]. CELLECT has been previously described[28]. Briefly, CELLECT generates genetic prioritization scores for each gene based on cell-type-expression specificity and GWAS summary statistics. As input to CELLECT, we used summary statistics derived from genetic studies of UK Biobank data for the presence of COPD[29] and lung function[30]. A complete list of genome-wide association summary statistics are available at the database of Genotypes and Phenotypes (dbGaP) under accession phs000179.v6.p2 for COPD (defined as FEV1/FVC ratio) and UK Biobank GWAS summary statistics for FEV$_1$/FVC ratio are available at www.ebi.ac.uk/gwas, study accession GCST007431. First, cellular expression-specificity scores were calculated for each gene using CELLEX [CELL-type EXpression-specificity] (v.1.2.1) using recommended normalization method and preprocessing steps (common transcript count normalization log-transformation). Using CELLEX, an expression-specificity likelihood (ESμ) was then computed for each gene across different cell types. We then integrated these results with summary statistics derived from genetic studies of UK Biobank data for the presence of COPD and lung function. In these studies, COPD was defined using pre-bronchodilator spirometry according to modified Global Initiative for Chronic Obstructive Lung Disease criteria. We ran CELLECT with the recommended workflow (CELLECT-LDSC) and default parameters (100 kb window size around each gene).

**Lung connectome to identify cell–cell interactions**. Methods to generate the lung connectome have been previously described[11]. Average expression values for every gene within cell types were calculated and mapped against the FANTOM5 database of known ligand–receptor pairs to create a global connectome using the R software Connectome (v0.2.2) (https://msraredon.github.io/Connectome/). Nodes were

defined as cellular clusters. A directed edge was created connecting two nodes if >5% of cells within the two cell types expressed the cognate molecules of the ligand–receptor pair. Directionality was therefore defined as outgoing signals from ligands and incoming signals to receptors. In general, edge weights represent the product of the average expression values of the ligand and receptor within their respective cell types, and cumulative edge weights are defined as the sum of the weights of all edges connecting pairs of cell types (i.e., nodes). Kleinberg hub and authority scores are metrics of outgoing and incoming centrality, respectively, that take into account the number and weights of edges connecting a node in a network; as such, a large hub score represents a node (cell cluster) that is highly connected or "central" to a network because it sends many outgoing signals with large edge weights and a large authority score represents a node that receives many incoming signals with large edge weights. Kleinberg hub and authority scores were calculated for each node using the igraph package in R. The connectome was filtered to edges between predefined cell types for the analysis. For Fig. 5B (and all pathway analyses comparing differences between COPD and control), we used scaled gene expression values to define the edge weights in which the weight of each edge was calculated as the product of average z-scores of the ligand and receptor within their respective clusters. A discussion about the use of scaled vs. unscaled expression values within the connectome analysis is further detailed by Raredon et al[11]. We plotted non-directional cumulative edge weights in the network graphs. For the pathway centrality analysis, the global connectome was filtered to ligand–receptor pairs that were preassigned to specific signaling modes. Edge weights were calculated using average unscaled gene expression values. Cumulative outgoing and incoming edge weights were computed for each node within every signaling mode; these values were scaled by mode and direction of signaling. Kleinberg hub and authority scores were also computed for each signaling mode. The Durbin test was used to assess global differences in signaling between control and COPD for each mode. A differential connectome was generated by computing the log-fold change of normalized expression values of the ligand and receptor for each edge between control and COPD. A perturbation score was then computed as the absolute value of the product of these fold changes for each edge. Ligand–receptor interactions for CXCL signaling were visualized with circos plots using the R package circlize after filtering the differential connectome based on the following criteria: both ligands and receptors being increased in COPD; ligands and receptors being expressed in at least 5% of the cells of their respective cell types; and omitting edges with perturbation scores less than 0.10. Ligand–receptor interactions for AT2 cells were visualized with circos plots based on the following criteria: Genes were differentially expressed between $AT2_S$ and $AT2_B$ based on Wilcoxon rank-sum test and Bonferroni-corrected $P < 0.05$; ligands and receptors being expressed in at least 5% of the cells of their respective cell types.

**RNA sequencing of isolated AT2 cells**. The isolation of AT2 cells from cryo-preserved single-cell suspensions of lung tissues samples used in this study has been previously described[15]. Briefly, cryopreserved single-cell suspensions of explanted lung tissue were obtained from 10 subjects with advanced COPD and 16 controls in the same manner as described for scRNAseq above. Cells were stained using the following antibodies or isotype-matched antibodies as negative controls: PE anti-human CD326 (EpCAM) (eBioscience 5011259, 1:100), FITC anti-human CD45 (BD bioscience 340664, 1:50), and Alexa Fluor® 647 anti-human podoplanin (PDPN) (BioLegend, 395003, 1:50), and DAPI (BioLegend) or propidium iodide. Cells were sorted using a BD FACSAria II (BD Biosciences). EpCAM^high/PDPN^− sorted cells were enriched for AT2 cell markers and used for further RNA sequencing. Library construction for RNA sequencing was performed as previously described using the Illumina TRuSeq RNA Access Library Prep kit (San Diego, CA) for library preparation, and sequencing was performed on a 75 bp paired-end flowcell using a HiSeq 2500 System. Each lane was spiked with 5% PhiX control libraries. Fastq files were trimmed using TrimGalore! (v0.6.6), before aligning to the human genome (GRCh38 p13) using the STAR aligner (2.7.5c). After alignment genes were quantified using featureCounts (v2.0.1). Samples were filtered to remove genes with low expression across all samples before normalization using the trimmed to means method. Differential gene expression was determined using generalized linear models, and after fitting differential expression was determined using a quasi-likelihood F-test. In the generalized linear models, the sequencing lane was treated as a blocking variable and the phenotype (COPD or Control) was treated as the main predictor.

**Lung genomics research consortium (LGRC) cohort**. Cyclic loess normalized NUPR1 expression was measured at the probe level (A_24_P270728) from the Affymetrix Human Gene 1.0 ST Array (Affymetrix) in 208 patients with COPD in the previously described LGRC cohort (GSE47460)[5,17].

**Cell culture**. A549 cells (American Type Culture Collection, #CCL-185) were cultured in DMEM supplemented with 10% heat-inactivated fetal bovine serum (FBS). The cells were passaged <20 times. Small airway epithelial cells (American Type Culture Collection, #PCS-301-010) were cultured in small airway epithelial growth media (Lonza #CC-3118) and passaged < 7 times. For studies using cell

death inhibitors, deferoxamine mesylate (Selleckchem #S5742), necrosulfonamide (Selleckchem #S8251), and Z-VAD-FMK (Selleckchem #S7023) were dissolved in DMSO, and cells were treated at indicated concentrations

**iPSC-derived AT2 cells grown at the air-liquid interface**. iPSC-derived AT2 cells at the air-liquid interface were generated as previously described.[36] Briefly, SFTPC^+ cells were derived from human iPSCs plated in Matrigel (Corning) droplets as described by Jacob et al.[78] Droplets were dissolved in dispase (Sigma) and alveo-lospheres were dissociated in 0.05% trypsin (Giboc) to generate single-cell suspensions. iPSC-derived AT2 cells were plated on Transwells in a solution containing 3 μM CHIR99021, 10 ng/mL KGF, 50 nM dexamethasone, 0.1 mM cAMP, 0.1 mM IBMX, 10 μM Rho-associated kinase inhibitor (Sigma Y-27632). 48 h later, apical media was aspirated, followed 24 h later by replacement of the media without Y-27632. Cells were dissociated from ALI using Accutase (Sigma) and viability was assessed using Zombie NIR Fixable Viability Kit (Biolegend #423105).

**NUPR1 siRNA**. RNA duplexes for silencing NUPR1 mRNA (siNUPR1) and non-targeting control were obtained from Dharmacon (L-012819-00-0005 ONTAR-GETplus Human NUPR1 (26471) siRNA- SMARTpool). Cells were transfected in 12-well plates using RNAiMAX transfection reagent (Life Technologies) and OptiMEM media according to the manufacturer's protocols. iPSC-derived AT2 cells were transfected with siRNA using RNAiMAX diluted in media and applied to the apical surface of the ALI for 24 h. Decreased NUPR1 expression was validated using the following primer sequences: Fwd, 5′-GGTCGCACCAAGAGAGAAGC-3′, Rev, 5′-CTCCGCAGTCCCGTCTCTAT-3′.

**Cigarette smoke extract (CSE)**. Mainstream smoke from one 3RF4 research cigarette (University of Kentucky, Lexington, Kentucky) was suctioned through 10 mL of cell culture media and filtered using a 0.22 μm filter (MilliporeSigma) as previously described, with the obtained filtrate considered 100% CSE[79]. Because CSE can vary batch to batch, we performed titration studies with each batch, using CSE concentrations that induced modest cell death in control cells.

**Flow cytometric measurements of cell death**. Flow cytometry for Annexin V-FITC and propidium iodide was performed per the manufacturer's protocol (BD Biosciences, 556547). For this cell death assay, we counted cells that were Annexin V-FITC positive and/or cells that were propidium iodide positive as a percent of total cells. Flow cytometry for caspase-3 and caspase-7 along with SYTOX AAD-vanced Dead Cell Stain was performed using CellEvent Caspase-3/7 Green Flow Cytometry Assay Kit per manufacturer's protocol (Thermo Fisher Scientific, C10427). Both analyses were performed using a Cytoflex LX flow cytometer and data was analyzed using Flow Jo 10.6 software.

**Cell viability and cytotoxicity assays**. Cell viability was assayed using MTT (3-(4,5-Dimethylthiazol-2-yl)-2,5-Diphenyltetrazolium Bromide) (Roche #1465007). In brief, cells were seeded into 96-well plates and incubated with the indicated treatments. Subsequently, 10 μL of MTT labelling reagent was added to the cells in a growth medium and incubated for 4 h (37 °C, 5% CO2), subsequently 100 μL of MTT solubilization solution was added and incubated overnight. Absorbance at 550 nm was measured the next day using a microplate reader (Molecular devices, vmax kinetic microplate reader). Cytotoxicity was assayed by using a lactate dehydrogenase (LDH)-cytox kit (Biolegend #426401). In brief, cells were seeded into 96-well plates and incubated with the indicated treatments. 50 μL of medium per well was transferred to a new 96 wells plate and 50 μL of the working solution was added to the medium. This was incubated for 1 h at 37 °C. Absorbance at 490 nm was measured using a microplate reader (Molecular devices, vmax kinetic microplate reader).

**Immunofluorescence for NUPR1, SFTPC, CD68, and MT2A**. Paraffin-embedded blocks of lung tissue samples were previously obtained from the Lung Tissue Research Consortium (LTRC). Control samples had no diagnosis of COPD and normal spirometry while our COPD samples were from subjects with GOLD stage III and GOLD stage IV disease and radiographic evidence of emphysema. For immunofluorescence staining, slides from paraffin-embedded blocks were depar-affinized in xylene and decreasing concentrations of ethanol in distilled water. They were then placed in EDTA pH 9 epitope retrieval buffer at 95 °C for 30 min then cooled, rinsed, and washed with TBS with 0.1% tween. Slides were incubated in a blocking buffer (Agilent) for 30 min. The primary antibodies applied overnight at 4 °C were: (1) mouse anti-SFTPC (sc-518029, Santa Cruz, 1:20); (2) rabbit anti-NUPR1 (ab234696, Abcam, 1:50); (3) mouse anti-MT2A, (MAB10176, R&D systems, 1:100); and (4) rabbit anti-CD68 (PA5-83940, Invitrogen, 1:100). Slides were washed with TBS with 0.1% tween and then incubated for 1 h at room temperature with secondary antibodies donkey anti-rabbit Alexa-488 (Thermo Fisher) and donkey anti-mouse Alexa-555 (Invitrogen). Slides were washed with TBS with 0.1% tween and the coverslip was mounted using anti-fade mounting media with DAPI (Vectashield). Images were acquired with a Nikon eclipse microscope.

**RNA in situ hybridization (RNA-ish) for *HHIP* and *CXCL12*.** Slides were deparaffinized in xylene and rehydrated with decreasing concentrations of ethanol in distilled water, and sections were treated with hydrogen peroxide (Advanced Cell Diagnostics (ACD), 322381) for 10 min at room temperature, and RNAScope 2.5 RED (ACD 322350) was used according to the manufacturer's instructions. Briefly, slides were incubated in 1× target retrieval reagent buffer (98–99 °C, 15 min) and treated with ProteasePlus (ACD 322330) for 25 min. Hybridization of target probes for *HHIP* mRNA (ACD464811) or *CXCL12* mRNA (ACD422991) and negative controls were performed according to the manufacturer's instructions in a HybEZ Oven (ACD). The Signal was visualized using Fast Red. For in situ hybridizations of *SFTPA1* mRNA and *HHIP* mRNA, together, we first hybridized with the target probe for *SFTPA1* mRNA (ACD891331) using BaseScope (ACD), and then we used the RNAscope® Fluorescent Multiplex Reagent Kit (ACD 320850) with AMP 4 A for detection of *HHIP* mRNA (ACD464811). Subsequent immunodetection of pro-SFTPC, PRX, and NOSTRIN was performed as previously described[80]. Briefly, after in situ hybridization, slides were washed and blocked in 10% goat serum in PBS and Triton X 0.3%. Incubation with primary antibody was performed overnight at 4 °C using: (1) rabbit anti proSPC (AB3786, Millipore, 1:500); (2) rabbit anti-PRX (NBP1-89598, Novus 1:400); and/or (3) mouse anti-NOSTRIN (sc373954, Santa Cruz 1:100). This was followed by detection with goat anti-rabbit-FITC and/or goat anti-mouse-AF647 antibody (Invitrogen, 31635 and Jackson Immuno Research, AB_2338902, respectively 1:500). To reduce autofluorescence, sections were incubated with TrueView reagent (Vectorlabs, SP-8500) for 3 min and mounted. Images were acquired with a Zeiss LSM710 confocal microscope equipped with a 20x objective.

**Quantitative immunofluorescence using CellProfiler.** CellProfiler pipelines were developed to quantify NUPR1 in AT2 cells and MT2A in CD68+ macrophages based on established protocols[81]. Briefly, all images were split into their respective channels. For channels measuring staining intensity of NUPR1 or MT2A, we applied CorrectIlluminationCalculate and CorrectIlluminationApply functions to compensate for illumination non-uniformities across images. In other channels, we applied RescaleIntensity. We then used IdentifyPrimaryObjects to identify AT2 cells or CD68+ cells. AT2 and CD68+ cells were filtered using MeasureObjectSizeShape and FilterObjects to filter cells of the appropriate size. We then used MaskImage and MeasureObjectIntensity to mask the image from the channel with NUPR1 or MT2A staining with the objects identified above. MeasureObjectIntensity was used to quantify staining within each object.

**Statistics.** Statistical analyses, adjustment for multiple comparison testing, biological replicates, and *P*-values are indicated in the results, methods, and figure legends. We used two-sided statistical tests for all comparisons. All error bars are defined in figure legends. No statistical methods were used to predetermine sample size. For mouse studies, littermates were randomly assigned to treatment groups.

**Reporting summary.** Further information on research design is available in the Nature Research Reporting Summary linked to this article.

## Data availability

The datasets generated during and/or analyzed during the current study are available or have been deposited on NCBI Gene Expression Omnibus. Specifically, Human scRNAseq data are available under accession code "GSE136831". Mouse scRNAseq is available under accession code "GSE168299". LGRC data are available under accession code "GSE47460". Human Single-cell expression data can also be interactively explored online at www.COPDcellatlas.com. A complete list of genome-wide association summary statistics are available at the database of Genotypes and Phenotypes (dbGaP) under accession phs000179.v6.p2 for COPD (defined as FEV1/FVC ratio) and UK Biobank GWAS summary statistics for FEV1/FVC ratio are available at www.ebi.ac.uk/gwas, study accession GCST007431. Mouse reference genome GRCm38 release M22 (GRCm38.p6) was downloaded from GENCODE (www.genecodegenes.org) and the human genome reference GRCh38 release 91 was downloaded from ENSEMBLE www.ensemble.org. All other relevant data supporting the key findings of this study are available within the article and its Supplementary Information files or from the corresponding author upon reasonable request. Source data are provided with this paper.

## Code availability

Data collection was performed with mkfastq pipeline in Cell Ranger's (v3.0.2), cutadapt (v1.17 and v2.9), zUMIs pipeline (v2.0), TrimGalore! (v0.6.6), STAR (v2.6.0c, v2.7.3a, and v2.7.5c). Single-cell analysis was performed using the Seurat R package (v3.2.3 and v4.0.4) using the recommended workflow. CELLEX (v1.2.1) CELLEX: (https://github.com/perslab/CELLEX) was performed using recommended normalization method and preprocessing steps. We then ran CELLECT v1.1.0 with the recommended workflow (CELLECT-LDSC) and default parameters (100 kb window size around each gene) (https://github.com/perslab/CELLECT). Connectome analysis was performed using R software Connectome (vl.0.0) https://msraredon.github.io/Connectome/). Average expression values for every gene within cell types were calculated and mapped against the FANTOM5 database of known ligand–receptor pairs to create a global connectome. For the centrality figure, the code we used was very similar to the CompareCentrality function in the Connectome (vl.0.0) package, but we changed the scaling method for visualization and added the Durbin significance test feature. The function for the network maps is not included in the Connectome package, so we added it. Finally, for the fold change calculations, the method was previously described[11], but the code was separate from the Connectome package. Modifications available to readers upon request. Other tools utilized include Fiji v1.0, GraphPad Prism v9.3.0, CellProfiler v4.2.1, featureCounts (v2.0.1), and UpsetR (v.1.4).

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

## Acknowledgements

This work was supported by NIH grants K08HL135402, R01HL155948, Flight Attendant Medical Research Institute, and support from the Claude D. Pepper Older Americans Independence Center at Yale School of Medicine (P30AG021342) to M.S. NIH grants R01HL127349, RO1HL141852, U01HL145567, UH3TR002445 to N.K, NHLBI P01 HL114501 and support from the Pulmonary Fibrosis Fund to I.O.R., an unrestricted gift from Three Lake Partners to I.O.R. and N.K., P01 HL152953-01A1 to A.A.W.; Department of Defense (W81XWH-19-1-0131) and German Research Foundation (SCHU 3147/1) to J.C.S., Austrian Science Fund (J4547) to T.B., F30 HL143906 to M.S.B.R., R01 HL138540 to L.E.N.; and an unrestricted research gift from Humacyte Inc. to L.E.N.

## Author contributions

M.S. and J.E.M. analyzed data, generated figures, and wrote the manuscript. T.S.A., J.C.S., N.Koth., C.Cos., and M.J.R. analyzed data. J.N., T.B., T.Y., M.C., R.B.W., T.Y., and N.O. performed experiments. S.P., E.A.E., and S.G.C. procured biospecimens and clinical data. K.H.J. and P.N.T. developed algorithms and performed data analysis. J.L.G. and C.J.B.

provided clinical insight and edited manuscript. M.S.B.R., L.E.N., K.H.J., C. Coa., A.A.W., and P.N.T. developed analytic approaches and/or supervised data analysis. N.Kam. and I.O.R. conceived project, supervised data analysis, and edited paper. All authors assisted in manuscript preparation and provided final approval of the submitted work.

## Competing interests

N.K. reports personal fees from Biogen Idec, Boehringer Ingelheim, Third Rock, Samumed, Numedii, Astra Zeneca, Life Max, Tervnce, RohBar, and Pliant. Equity in Pliant. Collaboration with Miragen, Astra Zeneca. Grant from Veracyte, all outside the submitted work; In addition, N.K. has a patent New Therapies in Pulmonary Fibrosis, and a patent for Peripheral Blood Gene Expression licensed to Biotech. L.E.N. reports grants from Humacyte Inc., outside the submitted work. E.A.A. and S.G.C. report personal fees from Novartis Institutes of Bio-Medical Research, outside the submitted work. K.H.J. and P.N.T. are employed by Intomics. The remaining authors declare no competing interests.

## Additional information

**Peer review information** *Nature Communications* thanks Ali Yildirim and the other anonymous reviewer(s) for their contribution to the peer review this work. Peer reviewer reports are available.

