## [Peer Review File · Nature Communications]

Reviewers' Comments:

Reviewer #1:

Remarks to the Author:

This manuscript used single-cell RNA-seq to characterize cell type specific changes in human COPD lungs. As it is a descriptive study and the scRNA-seq data has been published in their previous paper on IPF lungs, a thorough analysis of the data is needed for it to be impactful for the field.

Supplemental table 2 shows only a few hundred cells of the non-immune cell types were sequenced, which means ~20 cells per patient. Given the high dropout rate of 10X, this clear limitation should be acknowledged. A related point is to describe which regions of the lungs were processed for scRNA-seq and how this might contribute to inter-sample variations, which would be difficult to detect due to the small number of cells profiled. Please also add a column to supplemental table 2 to list the marker genes used for each cell type. It's also missing a row for aberrant basaloid cells.

AT2s was used to refer to a progenitor population based on the Travaglini et al paper. However, the letter "s" was used interchangeably for "stem" or "signaling". Signaling could refer to either an outgoing signal where the AT2 cells serve as a niche for other stem cells, or an incoming signal where AT2 cells receive a signal and react as stem cells. Please specify which genes predict AT2s as progenitors and different from AT2b, and discuss such genes in terms of outgoing and incoming signals. Feature plots of such genes, in comparison with that of SFTPC, are needed because the dot plot in Fig. 2C shows the generic AT2 gene SFTPC is similarly differential as the differential genes (e.g. WIF1 and HHIP). Related to predicting functions from gene expression, such hypothetical functions should be stated so when describing "Aerocytes uniquely express CA4 and HPGD and are specialized for gas exchange and leukocyte trafficking, while endothelial gCaps uniquely express FCN3 and promote vascular homeostasis by functioning as progenitor cells, modulating vasomotor tone, and regulating immune activation."

The comparison of GWAS candidates and scRNA-seq is interesting. To make the CELLECT analysis (Fig. 2E) informative, please include a table of the GWAS candidates enriched in each cell type and associated p-values. As HHIP, a strong GWAS candidate, is highlighted as unexpectedly expressed in AT2 cells, experimental validation by in situ, immunostaining, or RT-PCR (as in Fig. 3D) is needed.

The rationale in knocking down NUPR1 in smoke-exposed cells is confusing. Does smoke-exposure decrease NUPR1, as one might expect to mimic COPD? If so, why knocking it down further? If the hypothesis is that NUPR1 decrease somehow mediates the AT2 defects in COPD, the logical experiment would be to overexpress NUPR1 to see if it rescues the AT2 defects. Therefore, the baseline (i.e. control siRNA) levels of NUPR1 in this model and clear interpretation of the results are needed. On a technical note, PI and GFP could spectrally overlap giving false positives, so it is necessary to show the FMO controls. Is there an explanation for why PI displays a lower level of intensity in the last FACS plot in Fig. 3F? Also, the terms dead, susceptibility to death/apoptosis, and senescence seem to be used interchangeably and need clarification.

In Fig. 6, the UMAPs of control vs COPD lungs show a shift to the left in the COPD lungs. Could this be batch effect and thus confound the comparison of the relative abundance of different clusters? On the heatmap, MT2A is not unique to cluster 5 and has COPD-specific elevated expression in other macrophage clusters as well. Supporting this, while panel B shows cluster 5 is ~5% of the macrophages, the staining in panel D shows 2 out of 3 macrophages are positive for MT2A. The staining needs to be quantified, and the conclusion "we detected a metallothionein expressing macrophage subpopulation unique to COPD" likely needs modification.

Fig. 5B describes a strong outgoing signal from the gCap population and a strong incoming signal to the pDC population, indicating that these two cell types may be communicating. However, the human plot in panel C does not show much edge thickness between those two cell types. A comment on why this might be so is needed, and a list of CXCL pathway ligand-receptor pairs should be provided so others can reproduce the results.

Typo in line 329 "the marked increase in capillary endothelial inflammatory."

Reviewer #2:

Remarks to the Author:

This study delivers overall an important message that how COPD is heterogeneous and need to be consider many aspects for the understanding of underlying pathogenesis. Their findings are of interest, dissecting out the role ATII cells, capillary endothelial cells and macrophages plays in modulating COPD pathogenesis but I feel that the conclusions are often too strong for the data presented. For that reason, I would suggest that the following is undertaken to strengthen the existing data and support the conclusions made, before considering the manuscript for potential publication.

Fig 1. I believe as all used COPD patients had emphysema it is crucial to report emphysema development and inflammatory cell numbers (BALF) in CS exposed animals.

Fig 3. Your data is suggestive that AT2B cells are the cells expressing NURP1 in the lung. However, with the techniques readily available today I think it is imperative to undertake co-staining with Sftpc in human COPD and mouse lungs. Similarly, quantification of NURP1 needed from sections and/or protein evaluation should be included.

What is NUPR1 transcriptional regulator in smoker COPD but not in smoker control?

To support your conclusion that NURP1 deficiency increases cell death in cell lines, it would be value to repeat the primary AT2 cell experiments, in conjunction eg with NURP1 inhibitor. Is this caspase dependent cell death?

Fig 5. Could the author's present data showing for CXCL12 networks plot and in Fig 6 alveolar macrophage populations, particularly cluster 5, comparing "non-smoker control", "smoker control" and "smoker COPD" samples? This would bring out novel aspect of smoke induced COPD.

To strengthen the conclusions made in Fig 5C, I think it would be of value to stain human and lung tissue sectioning for CXCL12 with FCN3 or other gCap marker.

Similar to the other figures, I think you also have to reports macrophages population in CS mouse model.

Minor;

Supplemental Table 1. Do the authors know details about the smoking history?

Why they chose Sftpc Cre mTmG mice? Please describe.

Reviewer #3:

Remarks to the Author:

In this manuscript, Sauler and colleagues used single cell-transcriptome analysis to gain insights into COPD pathobiology. For this purpose, the authors reanalysed previously reported scRNA-seq datasets from control and COPD lungs. Additionally, the authors generated scRNA-seq data from cigarette smoke exposed mouse lungs, which they used to validate their conclusions from human COPD data analysis. In human lung epithelial scRNA-seq datasets, the authors identified two sub-populations of AT2 cells, which concurs with previous reports. It appears that At2-bulk cells, one of the two sub-population of AT2 cells is enriched and showed highest differentially expressed genes in COPD lung compared to controls. The authors claim that AT2-bulk cells exhibited altered metabolism and expresses stress response genes in COPD lungs compared to controls. The authors went on to knockdown NUPR1, one of the significantly altered gene in COPD, in A549 cell lines and claim that downregulation of NUPR1 results in a marked increase in cell death susceptibility in response to cigarette smoke extract in cultures. Furthermore, the authors characterized transcriptome in endothelial cells and macrophages in COPD lungs and found that

these cells have enhanced inflammatory and metallothioneine related gene expression signatures in COPD lungs compared to controls.

In summary, the authors made an attempt to further characterize human COPD lungs using previously reported scRNA-seq datasets and complemented them with newly generated scRNA-seq data from murine models of emphysema. The manuscript, however, lacks an in depth analysis of a specific population or a pathway that is altered in COPD. The authors need to buttress their findings derived from scRNA-seq data analysis with experimental evidence, a few of which are listed below.

Major comments:

1. Majority of Fig. 1 was already described in the authors previous studies (Adams et al., Science Advances, 2020).
2. SPA1 signal in Supplementary Fig. 3 does not appear to be specific. Additional markers and further analysis, such as isolation of subsets of these cells is needed to strengthen the claims. It appears that the lung tissue structure was not optimal in the image provided in this figure.
3. One of the key claim in this manuscript is that there is an enrichment of HHIP+ AT2 subpopulation in COPD. This requires additional marker analysis on tissue sections as well as functional analysis of the so called "AT2B" and AT2S cells authors can further strengthen their story by characterising the differences between AT2B and "AT2S" subpopulations. The authors could use ex vivo models of normal and COPD lungs to test the dynamics of these populations overtime following CSE treatment. Additionally, are these subpopulations similarly altered in other lung diseases or is it specific to COPD?
4. The authors used A549 cell line to determine the functional relevance of NUPR1 in the context of COPD. However, the author have not justified the use of A549 cells for this purpose. These are cancer cell lines and lack the cellular context (specifically AT2-Bulk vs AT2-S). The authors could use organoids derived from adult AT2 cells or iPSC-derived AT2 cells to validate their hypotheses derived from scRNA-seq data analysis.
5. The authors need to experimentally validate their observations on endothelial cells.
6. The authors need to explain the importance of metallothionein-rich macrophage sub-population in COPD.

Dear editor and reviewers,

Thank you for spending time to critically appraise our manuscript. We made substantial revisions and additions
to the manuscript based on your critical appraisal and we believe our manuscript is indeed better because of
your comments. A common theme from the review of our manuscript was the need for validation studies.
Therefore, we performed additional studies that validated key findings. Collectively our findings create a novel
depiction of the alveolus in COPD with cell-specific resolution. A summary of these findings and a description of
additional studies performed are listed below followed by a point-by-point response.

- 1) In the original version of our manuscript, we described a novel role for an alveolar epithelial type II
subpopulation (AT2_B cells) as a key cell type in COPD pathobiology. This newly described cell type had
the greatest degree of transcriptional aberrancy in COPD amongst all epithelial cells, and by integrating
our single-cell RNA sequencing findings with GWAS data, we found these cells to be key mediators of
COPD genetic susceptibility amongst epithelial cells. We also found AT2_B cells are the major expressors
of HHIP, a gene frequently implicated in COPD GWAS. **In our revised manuscript**, we have now
performed *in situ* hybridization studies validating the location of HHIP in AT2 cells, particularly in cells
that express SFTPA1 (an AT2_B cell marker). We further demonstrated that in FACS-isolated AT2 cells,
HHIP expression correlated with AT2_B markers including SFTPA1, SFTPA2, SFPTC, and NAPSA.
- 2) Our finding that COPD heritability is driven by genes expressed in structural cells, rather than
inflammatory cells, is novel and important in our understanding of COPD.
- 3) We provide a high-resolution characterization of AT2_B cells in COPD, including transcriptional evidence
for aberrant cellular metabolism and reduced antioxidant capacity. Additionally, we identified decreased
AT2_B expression of the stress-response gene NUPR1, a novel potential mechanism of COPD
pathogenesis. We independently validated decreased NUPR1 expression using isolated alveolar
epithelial cells from explanted lungs and mouse COPD models. **In our revised manuscript**, we further
validate decreased NUPR1 using quantitative immunofluorescence staining and performed functional
studies of *NUPR1*-deficient cells using induced pluripotent stem cells (iPSC)-derived AT2 cells grown at
air-liquid interface, primary small airway epithelial cells, and A549 cells, and all studies demonstrated that
decreased NUPR1 increased susceptibility to cell death.
- 4) Using our “connectome” analysis, we identified dysregulated cellular networks and signaling pathways,
thus identifying potential therapeutic targets in COPD. The largest identified change in cell-cell signaling
was outgoing CXCL signaling from general capillary endothelial cells (gCap), particularly CXCL12, which
was also identified in our mouse lung connectome. This finding was complemented by transcriptional
evidence for endothelial injury and capillary inflammation, suggesting a key role for endothelial cells as
mediators of COPD inflammation. **In our revised manuscript**, we validated capillary CXCL12 signaling
in COPD by performing *in situ* hybridization for *CXCL12* and co-stained with capillary and gCap cell
markers PRX and NOSTRIN.
- 5) We identified a novel population of macrophages that are high-expressors of metallothionein and HMOX1
in COPD. **In our revised manuscript**, we further validated the high-metallothionein expressing
macrophage population in COPD using quantitative immunohistochemistry.

Below is a point-by-point response to the specific remarks.

**Reviewer #1 (Remarks to the Author):** 49

**1.1) This manuscript used single-cell RNA-seq to characterize cell type specific changes in human COPD**
**lungs. As it is a descriptive study and the scRNA-seq data has been published in their previous paper**
**on IPF lungs, a thorough analysis of the data is needed for it to be impactful for the field.**

While some of the scRNAseq data has been published before, the previous study focused on idiopathic
pulmonary fibrosis and did not investigate differences between control and COPD samples. This manuscript
indeed describes a thorough analysis that specifically evaluates changes between control and COPD and

identifies novel findings that impact our understanding of COPD (as described above). These changes were not
described or identified in the previous manuscript. In addition to a detailed analysis of the original single cell
data set that identifies novel cellular and molecular features we include additional analyses and validation
studies, including studies in mice exposed to cigarette smoke, isolated AT2 cells from individuals with and without
COPD, quantitative immunofluorescence, *in situ* hybridization, and *in vitro* models of alveolar epithelial cells.

**1.2 Supplemental table 2 shows only a few hundred cells of the non-immune cell types were sequenced,**
**which means ~20 cells per patient. Given the high dropout rate of 10X, this clear limitation should be**
**acknowledged.**

We completely agree with the reviewer's comment that dropout rates and having too few cells are limitations of
this and other single cell studies. Therefore, we have now added a lengthier description to the Discussion
regarding these limitations. Specifically, we wrote:

*“There are also limitations common to scRNAseq experiments that deserve mention. These include “dropout” in*
*which transcripts, particularly from lowly expressed genes, are not detected, causing the data to be sparse and*
*zero-inflated. Consequently, our analyses are more likely to be biased towards detecting differences amongst*
*highly variable genes that are less affected by dropout. Another important limitation is dissociation bias due to*
*variable cellular sensitivities to digestive enzymes and differences in how cells are embedded in the extracellular*
*matrix. Such biases can lead to cellular proportions that are different from those found in vivo. As an example,*
*dissociation biases contributed to the sequencing of only 631 AT1 cells (average of 20 cells/subject) but 22,998*
*interstitial macrophages (average of 719 cells/subject), which we know isn't representative of what is in the lung.”*

It also deserves mention that this study remains one of the largest human lung single-cell studies to date, there
is no single subject with an overwhelming contribution to the data that biased the results, and we validated key
findings using multiple methods.

**1.3 A related point is to describe which regions of the lungs were processed for scRNA-seq and how this**
**might contribute to inter-sample variations, which would be difficult to detect due to the small number**
**of cells profiled.**

We appreciate this comment from the reviewer. Therefore, we have now added more details to the methods and
results section of the manuscript to demonstrate how the lung samples were obtained. Specifically, in the
Methods section we wrote:

*“Explanted organs were longitudinally sliced, and three to four longitudinal biopsies, containing tissue from*
*apical to basal segments of the lung, were washed with cold sterile phosphate-buffered saline (PBS) and visible*
*airway structures, vessels, blood clots, and mucin were removed. Lung tissue was mechanically minced,*
*enzymatically digested, and cryopreserved.”*

We have also added a statement to the discussion related to the role that sample processing may have on inter-
sample variations. Specifically, we wrote:

*“While all COPD subjects had radiographic emphysema, we do not know the degree of emphysema at the specific*
*location from where tissue was sampled and we cannot be sure that unrecognized systemic biases in sample*
*processing did not influence the results of our findings. However, samples were processed in a stereotyped fashion*
*in which lung samples were longitudinal sliced from apical to basal segments to minimize any sample processing*
*biases and biases caused by spatial heterogeneity.”*

**1.4) Please also add a column to supplemental table 2 to list the marker genes used for each cell type.**
**It's also missing a row for aberrant basaloid cells.**

Thank you for pointing this out. We have now added a list of marker genes for each cell, including aberrant
basaloid cells in Supplemental Table 2.

1.5) AT2s was used to refer to a progenitor population based on the Travaglini et al paper. However, the letter “s” was used interchangeably for “stem” or “signaling”. Signaling could refer to either an outgoing signal where the AT2 cells serve as a niche for other stem cells, or an incoming signal where AT2 cells receive a signal and react as stem cells. Please specify which genes predict AT2s as progenitors and different from AT2b, and discuss such genes in terms of outgoing and incoming signals.

We thank the reviewer for this comment and have modified the manuscript which includes now a better discussion of AT2_s and AT2_B cells in the text. We wrote:

“Among AT2 cells, we identified two cellular clusters that were recently described in the scRNAseq analysis of the normal human lung by Travaglini et al¹⁸. Both AT2 clusters expressed SFTPC, but one cluster had increased expression of “signaling” genes related to the role of AT2 cells as alveolar epithelial progenitors (AEPs) (hereon called AT2_s cells), while the other cluster express higher levels of canonical “bulk” AT2 cell markers (hereon called AT2_B cells)¹⁹⁻²¹ (Figure 2C, Supplemental Figure 3). We compared differentially expressed genes (DEGs) between the two AT2 cell populations, and found DEGs in AT2_s cells were enriched for gene ontology (GO) pathways related to cell differentiation (FDR = 5.89x10⁻⁶) and cell development (5.92x10⁻⁶) and had increased expression of genes associated with AEP function, such as EGF family receptor (ERBB4), WNT (TNIK, TCF12), and other developmental signaling pathways (FOXP1, STAT3, YAP, TEAD1). In contrast, DEGs in AT2_B cells were enriched for GO pathways related to metabolic processes (FDR = 4.18x10⁻³⁹) and biosynthetic processes (FDR = 2.64x10⁻³⁷) and included increased expression of classic AT2 markers such as SFTPA1, SFPTA2, and ETV5, and increased expression of WIF1, a negative regulator of WNT signaling.”

In the Discussion, we also wrote:

“AT2_B and AT2_s cells are subtypes of AT2 cells that we identified in our scRNAseq analysis of human lung tissue samples and were recently described by Travaglini et al.¹⁸ AT2_s cells had increased expression of genes related to cellular development and therefore may represent AEPs, while AT2_B cells had increased expression of genes related to surfactant production. Interestingly, this increase in genes implicated in surfactant production and other homeostatic AT2 functions was the largest difference between AT2_B and AT2_s cells, suggesting that contrasting properties of these two cell populations are not only related to AT2 progenitor function, and therefore may extend beyond the eponymous functions of AEPs and non-AEPs..”

Finally, we include a differential connectome of AT2_s and AT2_B cells in Supplemental Figure 17 These represent either “outgoing” ligands or “incoming” receptors that were differentially expressed (Bonferonni p<0.05) between AT2_B and AT2_s

Outgoing AT_B signaling:

Incoming AT_{2B} signaling:

1.6) Feature plots of such genes, in comparison with that of SFTPC, are needed because the dot plot in Fig. 2C shows the generic AT2 gene SFTPC is similarly differential as the differential genes (e.g. WIF1 and HHIP).

We agree that feature plots of key AT2 genes are a clear way of demonstrating differences in gene expression between AT_{2S} and AT_{2B} cells. While the size of such plots precludes them from being added to the main text, we have included them in **Supplemental Figure 2** (a smaller version of Supplemental Figure 2 is shown below).

**1.7) Related to predicting functions from gene expression, such hypothetical functions should be stated**
 **so when describing “Aerocytes uniquely express CA4 and HPGD and are specialized for gas exchange**
 **and leukocyte trafficking, while endothelial gCaps uniquely express FCN3 and promote vascular**
 **homeostasis by functioning as progenitor cells, modulating vasomotor tone, and regulating immune**
 **activation.”**

We have now included a discussion of hypothetical function when describing the AT2_B and AT2_S populations,
 and we also performed pathway enrichment for DEGs between AT2_B and AT2_S cells. We now state:

*“Among AT2 cells, we identified two cellular clusters that were also recently described in the scRNAseq analysis*
 *of the normal human lung by Travaglini et al¹⁸. Both AT2 clusters expressed SFTPC, but one cluster had increased*
 *expression of “signaling” genes related to the role of AT2 cells as alveolar epithelial progenitors (AEPs) (hereon*
 *called AT2_S cells), while the other cluster express higher levels of canonical “bulk” AT2 cell markers (hereon*
 *called AT2_B cells)¹⁹⁻²¹ (Figure 2C, Supplemental Figure 3). In comparing differentially expressed genes (DEGs)*
 *between the two AT2 cell populations, we found AT2_A cells were enriched in pathways related to cell*
 *differentiation (FDR = 5.89x10⁻⁶) and cell development (5.92x10⁻⁶) and had increased expression of genes*
 *associated with AEP function, such as EGF family receptor (ERBB4), WNT (TNIK, TCF12), and other*
 *developmental signaling pathways (FOXP1, STAT3, YAP, TEAD1) In contrast, DEGs in AT2_B cells were enriched*
 *for pathways related to metabolic processes (FDR = 4.18x10⁻³⁹) and biosynthetic processes (FDR = 2.64x10⁻³⁷)*
 *and included increased expression of classic AT2 markers such as SFTPA1, SFPTA2, and ETV5, and increased*
 *expression of WIF1, a negative regulator of WNT signaling.”*

**1.8) The comparison of GWAS candidates and scRNA-seq is interesting. To make the CELLECT analysis**
 **(Fig. 2E) informative, please include a table of the GWAS candidates enriched in each cell type and**
 **associated p-values.**

We agree that an understanding of top COPD GWAS genes would be informative to readers. Therefore, we
 have included a dot plot of z-scores for commonly associated GWAS genes in **Supplemental Figure 6** (smaller
 version shown below). A complete list of genome-wide association summary statistics are available at the
 database of Genotypes and Phenotypes (dbGaP) under accession phs000179.v6.p2 for COPD (defined as
 FEV₁/FVC ratio) and UK Biobank GWAS summary statistics for FEV₁/FVC ratio are available at
 www.ebi.ac.uk/gwas, study accession GCST007431. A summary of the expression specificity likelihood for each
 gene across different cell types is now included (**Supplemental Table 3**).

1.9) As HHIP, a strong GWAS candidate, is highlighted as unexpectedly expressed in AT2 cells, experimental validation by in situ, immunostaining, or RT-PCR (as in Fig. 3D) is needed.

We agree with the reviewers that the unexpected expression of HHIP in AT2 cells required experimental validation. We have done multiple experiments including *in situ* hybridization and quantification of HHIP in isolated AT2 cells as described in the text below and associated figures. We wrote:

“Of the cells expressing HHIP (>1 transcript), 80% were AT2_B cells (**Supplemental Figure 4**). To validate these findings, we performed *in situ* hybridization (**Figure 2D**) and quantified HHIP in SFTPC⁺ cells. We observed 48% of SFTPC⁺ cells contained HHIP foci and 79% of all HHIP foci co-localized with SFTPC⁺ cells. We then assessed HHIP expression in isolated human AT2 epithelial cells obtained by fluorescence-activated cell (FACS) sorting single-cell suspensions of human lung tissue samples (EpCAM^{high}/PDPN⁻)¹⁵. We found HHIP expression correlated with the expression of AT2_B cell markers identified independently through scRNAseq including: SFTPA1 ($\rho = 0.94$), NAPSA ($\rho = 0.91$), CA2 ($\rho = 0.91$), SFTA2 ($\rho = 0.90$), LAMP3 ($\rho = 0.90$), SFTPC ($\rho = 0.90$), SFTPD ($\rho = 0.89$), SFTPA2 ($\rho = 0.88$), WIF1 ($\rho = 0.88$), PGC ($\rho = 0.86$), and ETV5 ($\rho = 0.85$) (Bonferroni-corrected p -value <0.01) (**Figure 2E**). Because SFTPA1 expression was greater in AT2_B cells and expression of HHIP and SFTPA1 in FACS-isolated AT2 cells were highly correlated, we performed *in situ* hybridization for HHIP, SFTPA1, and immunostaining for SFTPC (**Supplemental Figure 5**), and confirmed HHIP co-localized with SFTPA1 in a subset of SFTPC⁺ cells.”

From Supplemental Figure 4

From Figure 2D
SFTPC/HHIP/DAPI

From Figure 2E

From Supplemental Figure 5

1.10) The rationale in knocking down NUPR1 in smoke-exposed cells is confusing. Does smoke-exposure decrease NUPR1, as one might expect to mimic COPD? If so, why knocking it down further? If the hypothesis is that NUPR1 decrease somehow mediates the AT2 defects in COPD, the logical experiment would be to overexpress NUPR1 to see if it rescues the AT2 defects. Therefore, the baseline (i.e. control siRNA) levels of NUPR1 in this model and clear interpretation of the results are needed.

We appreciate the reviewer's comments, and apologize for the confusion. It is not uncommon that a stress induced gene is decreased in a disease condition, potentially reflecting a failed compensatory response or an individual predisposition to injury. We have conducted multiple experiments to better understand the role of NUPR1 in COPD. First, we find exposure to cigarette smoke extract (CSE) increases NUPR1 expression, similar to previous studies demonstrating NUPR1 to be a stress-inducible gene (Supplemental Figure 9), and consistent with this finding, we found AT2_B expression of NUPR1 was increased in smokers without COPD compared to never smokers without COPD (Figure 3B). Because our data suggested NUPR1 increases in response to CS in healthy individuals, but is decreased in COPD, we hypothesized impaired NUPR1 responses may increase susceptibility to cigarette smoke driven injury. We tested our hypothesis by inhibiting NUPR1 using siRNA in multiple model systems including A549 cells, small airway epithelial cells, and iPSC-derived AT2 cells grown at air liquid interface, and exposing cells to CSE. Knockdown is shown in Supplemental Figure 10. In all three models, inhibition of NUPR1 increased susceptibility to cell death in response to injury, highlighting the potential epithelial protective role of NUPR1, and the potential role of its inhibition in the development of COPD (Figure 3H&J). While studying in detail the effects of NUPR1 was beyond the scope of this study, we assessed the type of cell death potentially driven by loss of NUPR1. We found that inhibition of NUPR1 increases susceptibility to cell death, possibly through ferroptosis; Caspase-dependent cell death was not a major consequence of NUPR1 deficiency in A549 cells (Supplemental Figure 13) and a screen of various cell death inhibitors (Supplemental Figure 14) revealed that deferoxamine mesylate reversed susceptibility to cell death, a finding that we further validated using flow-cytometric detection of Annexin V and propidium iodide (Figure 3I,J). This is in agreement with recent studies (Yoshida, Nature Communications, 2019) that have implicated ferroptosis as a mechanism of cell death in COPD. We have modified the discussion to reflect these points. While this is obviously outside the scope of this manuscript, we plan to pursue a better understanding of the relationship between NUPR1 and ferroptosis in the future. Finally, we did try overexpressing NUPR1 in A549 to see its effect on survival. Notably, overexpressing NUPR1 decreased cell death at baseline. However, there was no difference in cell death in response to CSE between cells transfected with vehicle control and cells transfected with the NUPR1 overexpression vector. Collectively, overexpression of NUPR1 may not reduce CSE-mediated cell death because NUPR1 is already increased in response to CSE. Additionally, there are many

examples of cellular stress responses that when overexpressed have no beneficial effect. For example, overexpression of DNA repair proteins does not improve DNA repair in healthy cells. There is clearly much to be worked out regarding the role of NUPR1 in lung epithelial cell death, but we believe that a mechanistic study of NUPR1 in COPD is beyond the scope of this manuscript.

From Supplemental Figure 9

From Supplemental Figure 10

From Figure 3H:

From Figure 3J:

From Supplemental Figure 13

NUPR1 overexpression

**1.11) On a technical note, PI and GFP could spectrally overlap giving false positives, so it is necessary**
**to show the FMO controls.**

We agree with the reviewers. Therefore, we have now included FMO controls for flow-cytometry in
**Supplemental Figures 11, 12, and 13.** In addition, it is important to mention that the results of flow cytometric
study are not subtle. Inhibition of *NUPR1* increased susceptibility to CSE-mediated cell death in iPSC-derived
AT2 cells (fold change = 1.5), primary small airway epithelial cells (fold change = 1.9), and A549 cells (fold
change = 21.3). Therefore, within the range that gates can be set as dictated by our FMO controls, inhibition of
*NUPR1* increases susceptibility to CSE-mediated cell death.

**1.12) Is there an explanation for why PI displays a lower level of intensity in the last FACS plot in Fig.**
**3F? Also, the terms dead, susceptibility to death/apoptosis, and senescence seem to be used**
**interchangeably and need clarification.**

We were also intrigued by this finding. Therefore, we sought to identify the cell-death pathway(s) that might be
contributing to increased susceptibility to cell death. Please see response to **reviewer 1 question 1.10** above.
Briefly, we first assessed CSE-mediated cell death using a caspase 3/7 assay and found that cell death was
largely caspase 3/7-independent (**Supplemental Figure 13**). We then screened various inhibitors of ferroptosis
(deferrioxamine), necroptosis, necrosulfonamide (MLKL inhibitor) and Z-VAD-FMK (pan-caspase inhibitor)
(**Supplemental Figure 14**). We found that deferrioxamine reversed the increase in cell death caused by *NUPR1*
inhibition. We now discuss the potential role of *NUPR1* and ferroptosis in the context of COPD. We write:

*“Aberrant iron metabolism and ferroptosis have been recently implicated in COPD pathogenesis.^{47,55} Ferroptosis*
*is also associated with decreased antioxidant capacity and the accumulation of lipid reactive oxygen species*
*(ROS).⁵⁶ Therefore, decreased AT2B expression of NUPR1 coupled with aberrant AT2B bioenergetics may*
*increase susceptibility to ferroptosis in these cells. Future studies will be necessary to dissect the complex*
*interplay between oxidative stress, NUPR1, and iron metabolism in COPD.”*

**1.13) In Fig. 6, the UMAPs of control vs COPD lungs show a shift to the left in the COPD lungs. Could**
**this be batch effect and thus confound the comparison of the relative abundance of different clusters?**

We agree with the reviewer that batch effect can have an outsized influence on clustering and interpretation of
the data, but we do not think that this is the case. In our study, all samples were cryopreserved after isolation,
and sequencing was performed using a block design in which equal numbers of control and disease samples
were run at the same time. Shifts within-cluster (i.e. local) populations from UMAP images may be over-
interpreted and not represent significant shifts in expression within the cluster (ref:
<https://www.biorxiv.org/content/10.1101/2021.08.25.457696v3>). Last, clusters were defined by marker genes
for canonical cell types and we didn't note any misidentified cell types because of batch or disease phenotype.
Overall, we do not believe that batch contributes to identification of cell types or relative abundance
measurements in our study.

**1.14) On the heatmap, MT2A is not unique to cluster 5 and has COPD-specific elevated expression in**
**other macrophage clusters as well. Supporting this, while panel B shows cluster 5 is ~5% of the**
**macrophages, the staining in panel D shows 2 out of 3 macrophages are positive for MT2A. The staining**
**needs to be quantified, and the conclusion “we detected a metallothionein expressing macrophage**
**subpopulation unique to COPD” likely needs modification.**

We agree with the reviewer that our conclusion requires modification and should be more nuanced. While MT2A
MT2A is not unique to cluster 5, MT2A is generally increased in COPD and MT2A expression is markedly
elevated in cluster 5 compared to other clusters. These data suggested to us that COPD was associated with a
general increase in MT2A expression and there was a “high-expressing” MT2A population that only occurred in
COPD. We performed quantitative immunostaining for MT2A in COPD, as suggested by the reviewer, which
further supported this hypothesis (**Figure 6E-G**). Therefore, we have restated our conclusion and write “elevated
metallothionein expression in a subpopulation of cells is observed in COPD.”

**1.15) Fig. 5B describes a strong outgoing signal from the gCap population and a strong incoming signal**
**to the pDC population, indicating that these two cell types may be communicating. However, the human**
**plot in panel C does not show much edge thickness between those two cell types. A comment on why**
**this might be so is needed,**

We agree with the reviewer that this may be confusing. In **Figure 5A**, we are showing the connectome that
represents all cell-cell ligand-receptor interactions, not just CXCL-pathway. In **Figure 5B**, we are specifically
showing changes in CXCL signaling. CXCL signaling is only one of many signaling pathways (highlighted in
**Supplemental Figure 18, Supplemental Table 4**). Therefore, we have tried to provide more clarity. We have
re-written the results section and write:

*“Each node represents a cell population, inter-nodal edges reflect nondirected ligand-receptor interactions, and*
*the edge weight, reflected by the thickness of the edge in our network map, represents the sum of all ligand-*
*receptor interactions between two nodes that have been identified in the FANTOM5 database... To further explore*
*changes in cellular communication, we evaluated ligand-receptor interactions that were preassigned to canonical*
*signalling pathways, such as WNT, Sempahorins, and PDGF; thus allowing us to identify changes in specific*
*cell-cell signaling pathways between control and COPD.”*

**1.16) and a list of CXCL pathway ligand-receptor pairs should be provided so others can reproduce the**
**results.**

We agree that including ligand-receptor pairs can improve the readability of the manuscript. Therefore, we have
now included a table of CXCL ligand-receptor pairs in **Supplemental Table 5**.

**1.17) Typo in line 329 “the marked increase in capillary endothelial inflammatory.”**

Thank you for pointing this out. We have now fixed this error.

**Reviewer #2 (Remarks to the Author):**

**2.1) This study delivers overall an important message that how COPD is heterogeneous and need to be**
**consider many aspects for the understanding of underlying pathogenesis. Their findings are of interest,**
**dissecting out the role ATII cells, capillary endothelial cells and macrophages plays in modulating COPD**
**pathogenesis but I feel that the conclusions are often too strong for the data presented. For that reason,**
**I would suggest that the following is undertaken to strengthen the existing data and support the**
**conclusions made, before considering the manuscript for potential publication.**

We appreciate the reviewer’s interest in our study. We agree that this manuscript delivers an important message,
and in response to the reviewers’ comments, we have restated certain conclusions while also performing
additional validation experiments to bolster other conclusions.

**2.2) Fig 1. I believe as all used COPD patients had emphysema it is crucial to report emphysema**
**development and inflammatory cell numbers (BALF) in CS exposed animals.**

We now included measurements of emphysema for our animal models by reporting mean linear intercept
(**Supplemental Figure 2A**). We did not perform bronchoalveolar lavage on our mouse lungs because we did
not want to deplete the lungs of any specific cell population prior to single-cell RNA sequencing. However, our
analyses does suggest an increase in inflammatory cells in the lungs, because we quantified CD45+ cells
obtained via MACS sorting which were increased in cigarette smoke-exposed lungs (**Supplemental Figure 2B**).

2.3) Fig 3. Your data is suggestive that AT2B cells are the cells expressing NURP1 in the lung. However, with the techniques readily available today I think it is imperative to undertake co-staining with Sftpc in human COPD and mouse lungs. Similarly, quantification of NURP1 needed from sections and/or protein evaluation should be included.

We agree with the reviewer that co-staining was necessary to quantify NURP1 in AT2 cells. Therefore, we performed immunofluorescence staining of NURP1 in AT2 cells and quantified AT2-specific NURP1 in these cells in **Figure 3**. We found decreased NURP1 in COPD AT2 cells. These findings are in addition to decreased NURP1 identified in human AT2 cells using scRNAseq data, mouse AT2 cells using scRNAseq, and FACS-sorted isolated AT2 cells. Below is sample immunofluorescence staining, and the results from quantitative immunofluorescence of NURP1 in COPD.

From Figure 3F, G

**2.4) What is NUPR1 transcriptional regulator in smoker COPD but not in smoker control?**

We agree with the reviewer that understanding the regulatory mechanisms underlying decreased NUPR1 in
COPD is important. In a recent study by Morrow et al. (AJRCCM 2018), the investigators integrated findings from
DNA methylation profiling of human lung tissue and GWAS studies. They tested for SNPs associated with DNA
methylation levels, and then co-localized those findings to results from COPD GWAS to identify potential regions
where genetic polymorphism mediate their effects through aberrant epigenetic regulation. These investigators
identified differential methylation near NUPR1. Another study demonstrated that the extracellular matrix in COPD
can also decrease epithelial NUPR1 expression. Therefore, we have referenced these articles and provided a
more thorough commentary in our discussion. However, we also believe that understanding the transcriptional
regulation of NUPR1 is beyond the scope of this study. We write in the Discussion:

*“While the role of NUPR1 in non-malignant lung disease remains uncertain, an association between NUPR1 and*
*COPD was recently suggested by Morrow et al. who integrated findings from whole-genome methylation profiling*
*of human lung tissue with COPD GWAS.⁵³ They colocalized single-nucleotide polymorphisms (SNPs) identified*
*in COPD GWAS with local cis-regulation of CpG methylation, and suggested that SNPs in close proximity to*
*NUPR1 mediate local epigenetic regulation with relevance to COPD. Additionally, analyses by Hedstrom et al.*
*suggest extracellular matrix in COPD may exert diverse effects including decreased NUPR1 expression.⁵⁴”*

**2.5) To support your conclusion that NURP1 deficiency increases cell death in cell lines, it would be**
**value to repeat the primary AT2 cell experiments, in conjunction e.g. with NURP1 inhibitor. Is this**
**caspase dependent cell death?**

We agree with the reviewer comments and have undertaken extensive experimental testing to address this
concern, **detailed above in response to reviewer 1, comment 1.10**. Briefly, we inhibited NUPR1 using siRNA
in multiple model systems including A549 cells, small airway epithelial cells, and iPSC-derived AT2 cells grown
at air liquid interface. Knockdown is shown in **Supplemental Figure 10**. In all three models, inhibition of NUPR1
increased susceptibility to cell death (**Figure 3H&J**). We found that caspase-dependent cell death was not a
major consequence of NUPR1 deficiency in A549 cells (**Supplemental Figure 13**). Therefore, we screened
various cell death inhibitors (**Supplemental Figure 14**) and found deferoxamine mesylate reversed susceptibility
to cell death, a finding that we further validated using flow-cytometric detection of Annexin V and propidium
iodide (**Figure 3I,J**). Collectively, these data demonstrate inhibition of NUPR1 increases susceptibility to cell
death and suggests this may be through ferroptosis.

**2.6) Fig 5. Could the author’s present data showing for CXCL12 networks plot and in Fig 6 alveolar**
**macrophage populations, particularly cluster 5, comparing “non-smoker control”, “smoker control” and**
**“smoker COPD” samples? This would bring out novel aspect of smoke induced COPD.**

Yes. We have now included plots for CXCL12 networks and alveolar macrophage populations comparing “non-
smoker control”, “smoker control” and “smoker COPD” samples in **Supplemental Figures 22 and 24**. These
data suggest the increase in CXCL12 and metallothionein is specific to COPD and not simply due to cigarette
smoke exposure. However, we have chosen to put this in the supplement for the following reasons. First, we did
not identify plasmacytoid dendritic cells in current/former smokers, likely due to a small number of samples.
Therefore, whether the change in CXCL12 is due to smoking or specific to COPD requires further study.
Similarly, there remains an increase in high-metallothionein expressing macrophages exclusive to COPD, but
the number of current/former smokers without COPD is only 4, and therefore, we are including this in the
supplement for similar reasons.

**2.7) To strengthen the conclusions made in Fig 5C, I think it would be of value to stain human and lung**
**tissue sectioning for CXCL12 with FCN3 or other gCap marker.**

Thank you for this suggestion. We unsuccessfully attempted *in situ* hybridization for FCN3. Therefore, we
identified two additional markers to validate our findings. We performed *in situ* hybridization for CXCL12 and
PRX, a well-established capillary marker (**Supplemental Figure 15A**) (see below), as well as CXCL12 and
NOSTRIN, a nuclear protein expressed predominately in gCaps compared to aerocytes as shown in **in**

Supplemental Figure 15B (see below). Images showing localization of CXCL12 in PRX⁺ and NOSTRIN⁺ cells are shown below and have been included in **Figure 5C** (see below).

From Supplemental Figure 15

From Figure 5C

**2.8) Similar to the other figures, I think you also have to reports macrophages population in CS mouse**
**model.**

Single-cell studies of the lung are highly enriched for immune cells, particularly macrophages, because they are
hardy cells that are easily liberated from lung tissue. However, our findings were focused on non-immune cells.
Therefore, we performed CD45+ depletion of digested mouse lungs to enrich for non-immune cells in our murine
studies and confirm findings in humans, with the goal that 10% of cells were derived from the CD45+ population.
However, within this population we did not identify MT2A high-expressing macrophages, which may be a
consequence of antibody binding to macrophages, and the process of CD45-depletion. Therefore, we have not
included any further data about BAL macrophages from our mouse study. We also reanalyzed a publicly
available dataset of macrophages isolated from the BAL of smokers with COPD (GSE130928), published in the
article "Alveolar Macrophage Immunometabolism and Lung Function Impairment in Smoking and Chronic
Obstructive Pulmonary Disease, AJRCCM 2020," which also demonstrated an increase in metallothionein.
However, because we lacked key details about study subjects, we chose not to include this in our manuscript.

**Minor:**

**Supplemental Table 1. Do the authors know details about the smoking history?**

We have now included detailed smoking history in Supplemental Table 1.

**Why they chose Sftpc Cre mTmG mice? Please describe.**

We chose SFPTC-Cre mTmG mice because we were focused on understanding changes in AT2 cells. We have
now included mention of this in the results section. Specifically, we write " *To validate findings, particularly those*
*in AT2 cells, we performed scRNAseq of isolated lung cells from SftpcCre^{ERT2}-mTmG (AT2 cell) reporter mice.*"

**Reviewer #3 (Remarks to the Author) In this manuscript, Sauler and colleagues used single cell-**
**transcriptome analysis to gain insights into COPD pathobiology. For this purpose, the authors**
**reanalysed previously reported scRNA-seq datasets from control and COPD lungs. Additionally, the**
**authors generated scRNA-seq data from cigarette smoke exposed mouse lungs, which they used to**
**validate their conclusions from human COPD data analysis. In human lung epithelial scRNA-seq**
**datasets, the authors identified two sub-populations of AT2 cells, which concurs with previous reports.**
**It appears that At2-bulk cells, one of the two sub-population of AT2 cells is enriched and showed highest**
**differentially expressed genes in COPD lung compared to controls. The authors claim that AT2-bulk cells**
**exhibited altered metabolism and expresses stress response genes in COPD lungs compared to**
**controls. The authors went on to knockdown NUPR1, one of the significantly altered gene in COPD, in**
**A549 cell lines and claim that downregulation of NUPR1 results in a marked increase in cell death**
**susceptibility in response to cigarette smoke extract in cultures. Furthermore, the authors characterized**
**transcriptome in endothelial cells and macrophages in COPD lungs and found that these cells have**
**enhanced inflammatory and metallothionein related gene expression signatures in COPD lungs**
**compared to controls. In summary, the authors made an attempt to further characterize human COPD**
**lungs using previously reported scRNA-seq datasets and complemented them with newly generated**
**scRNA-seq data from murine models of emphysema. The manuscript, however, lacks an in-depth**
**analysis of a specific population or a pathway that is altered in COPD. The authors need to buttress their**
**findings derived from scRNA-seq data analysis with experimental evidence, a few of which are listed**
**below.**

**Major comments:**

**3.1. Majority of Fig. 1 was already described in the authors previous studies (Adams et al., Science**
**Advances, 2020).**

We appreciate the author's point about Figure 1. However, there are some key reasons why we included this
data. First, the data is not the same as we used samples that were matched for age and gender, and therefore
the population chosen for this experiment was slightly different. Additionally, we wanted to present a summary
of the data in order to improve readability of the text. Finally, while some of the scRNAseq data has been
published before, the previous study focused on comparing idiopathic pulmonary fibrosis with controls and did

not investigate differences between control and COPD samples. This manuscript specifically evaluates changes
between control and COPD and identifies novel findings that impact our understanding of COPD (as described
above) that were not identified in the previous manuscript (see response to **reviewer 1, question 1.1**). We now
include additional analyses and validation studies, including studies in mice exposed to cigarette smoke, isolated
AT2 cells from individuals with and without COPD, quantitative immunofluorescence, *in situ* hybridization, and
*in vitro* models of alveolar epithelial cells.

**3.2. SPA1 signal in Supplementary Fig. 3 does not appear to be specific. Additional markers and further**
**analysis, such as isolation of subsets of these cells is needed to strengthen the claims. It appears that**
**the lung tissue structure was not optimal in the image provided in this figure.**

We appreciate the reviewer's comments regarding *SFTPA1 in situ* hybridization. Therefore, we performed the
following experiments which are fully detailed in response to (**reviewer 1 section 1.9**) First, as the reviewer
suggested, we evaluated the expression of SFTPA1 in isolated AT2 cells, and we found that SFTPA1 along with
other AT2_B markers correlate with HHIP expression, (**Figure 2E**), providing independent validation of the AT2_B
cluster. We also repeated our *in situ* hybridization that shows co-localization of *SFTPA1* and *HHIP* in a subset
of SFTPC+ AT2 cells (**Supplemental Figure 5**) (also in **response to reviewer 1, section 1.9**). Finally, analysis
of data from independent studies by Travaglini *et al*, Nature 2020 and Zacharias *et al*, Nature 2018 demonstrate
that non-AEP cells are enriched for the expression of surfactants, similar to our reported findings.

**3.3a. One of the key claims in this manuscript is that there is an enrichment of HHIP+ AT2 subpopulation**
**in COPD.**

We appreciate the reviewer's interest in HHIP+ AT2 cells and the need for further analysis. However, we do not
claim that there is enrichment of HHIP+ AT2 cells in COPD, as the numbers of these cells between COPD and
control are not different. Our study identified important changes in gene expression within these populations that
may contribute to disease susceptibility.

**3.3b. This requires additional marker analysis on tissue sections as well as functional analysis of the so**
**called "AT2B" and AT2S cells authors can further strengthen their story by characterizing the**
**differences between AT2B and "AT2S" subpopulations. The authors could use ex vivo models of normal**
**and COPD lungs to test the dynamics of these populations overtime following CSE treatment.**

We agree with the reviewer that further validation was required. We have now performed *in situ* hybridization
for *HHIP* in AT2 cells as described in response to **reviewer 1 section 1.9**), including *in situ* hybridization for
*HHIP* and *SFTPA1*. We further demonstrated that in FACS-isolated AT2 cells, HHIP expression correlated with
AT2_B markers including SFTPA1, SFTPA2, SFPTC, and NAPSA. Therefore, our independent validation studies
plus our scRNAseq findings. Additionally, AT2_B and AT2_S populations were recently described by Travaglini *et al*,
and therefore our data set represents a second independent validation. We agree with the reviewer that
characterizing these populations in PCLS or other ex-vivo model over time will certainly be exciting, it is beyond
the scope of this paper. We actually plan on establishing a new dynamic model of AT2 cells in human PCLS and
validating this novel model in the future.

**3.4. Additionally, are these subpopulations similarly altered in other lung diseases or is it specific to**
**COPD?**

We evaluated the AT2 subpopulations in IPF as well. We found 43% of AT2 cells were AT2_B in control patients,
40% of AT2 cells were AT2_B in COPD patients, and 56% of AT2 cells were AT2_B in IPF patients. In examining
gene expression changes, the reduction of NUPR1, which was the most significant differentially expressed gene
in AT2_B cells and the focus of much of our AT2 cell analysis, was unique to COPD and showed no change in
AT2_B cells between controls and IPF.

**3.5. The authors used A549 cell line to determine the functional relevance of NUPR1 in the context of**
**COPD. However, the author have not justified the use of A549 cells for this purpose. These are cancer**
**cell lines and lack the cellular context (specifically AT2-Bulk vs AT2-S). The authors could use organoids**

**derived from adult AT2 cells or iPSC-derived AT2 cells to validate their hypotheses derived from scRNA-**
**seq data analysis.**

While A549 cells have long-been used as model for type 2 cells (Foster, Exp Cell Res 1998), we completely
agree with the reviewer they are not an ideal model because they are a cancer cell line and therefore their cellular
responses to stress and cell death responses are not the same as non-transformed cells. Therefore, we have
taken the reviewer's excellent advice and performed additional experiments where we inhibited NUPR1 using
siRNA in multiple model systems including newly described induced pluripotent stem cells (iPSC)-derived AT2
cells grown at air-liquid interface and primary small airway epithelial cells (SAECs) in submerged culture.
Knockdown is shown in **Supplemental Figure 10**. Inhibition of NUPR1 increased susceptibility to cell death
following exposure to CSE (**Figure 3H**). Details of our findings are described above (**response to reviewer 1,**
**section 1.10**).

**3.6. The authors need to experimentally validate their observations on endothelial cells.**

We now provide validation by performing CXCL12 *in situ* hybridization in endothelial cells in **Figure 5C**. We find
that in COPD, CXCL12 can be co-localized with lung capillary and gCap markers. A detailed description of
CXCL12 *in situ* hybridization is described above (**response to reviewer 2, section 2.7**). This is in addition to
our data demonstrating increased CXCL12 gCap signaling in mice exposed to 10 months of CS, (**Figure 5C,**
**Supplemental Figure 20**), providing further validation of increased capillary CXCL12 signaling. We considered
functional studies in endothelial cells, but recent studies have demonstrated that human lung endothelial cells
grown *in vitro* lose their defining transcriptional signatures. i.e. the transcriptional features that define these cells
as gCap, arterial, and aerocytes are lost (Schupp, Circulation, 2021). Additionally, we write in the Discussion:

*“Increased CXCL chemokine signaling has been previously implicated in lymphoid neogenesis and COPD*
*pathogenesis⁶², and a recent study demonstrated that AMD3100, a drug that inhibits the CXCL12 receptor*
*CXCR4, ameliorates CS-induced emphysema in a mouse model⁶³. Prior single-cell analyses have shown*
*pulmonary arteries, but not pulmonary capillary, express CXCL12 in healthy human lung tissue.⁶⁴ However in*
*COPD, our study prioritized gCap cells as a major source of outgoing endothelial CXCL12 signaling, a finding*
*we also demonstrated in mice and validated through in situ hybridization of CXCL12 in COPD lung tissue.”*

**3.7. The authors need to explain the importance of metallothionein-rich macrophage sub-population in** 709 **COPD.**

We agree with the reviewer that the importance of a metallothionein-rich macrophage sub-population needs
further clarification. We specifically focused on this macrophage population because it was the one cluster that
was enriched in COPD after multiple-comparison testing. We then validated metallothionein-rich macrophages
in COPD using quantitative immunofluorescence staining shown in **Figure 6 D-F (shown below)**. While the
specific role for metallothionein-rich macrophages remain unknown, we do include a discussion about what its
role might be. Specifically, we wrote in the Discussion:

*“Previous studies have shown that metallothioneins are induced by oxidative stress and inflammation, and protect*
*against cellular injury by sequestering intracellular metals such as zinc and copper⁶¹. Metallothionein has a*
*protective role in acute lung injury models⁶², but little is known about the consequences of chronic metallothionein*
*upregulation in the setting of advanced COPD where redox homeostasis and heavy metal metabolism are*
*disrupted.⁶³ However, aberrant metabolism of intracellular metals are already implicated in COPD pathogenesis.*
*For instance, Menkes diseases, a congenital disorder of copper deficiency, causes emphysema⁶⁴. Dysregulated*
*zinc homeostasis can cause impaired phagocytosis and an abnormal inflammatory response in macrophages⁶⁵.*
*Therefore, understanding metallothionein regulation of zinc and copper in COPD may improve our*
*understanding of disease pathogenesis.”*

From Figure 6D-F.

Reviewers' Comments:

Reviewer #1:

Remarks to the Author:

The revised manuscript is much improved. Two points need to be clarified in text.

First, the abstract says "Network analyses identified an important role for inflamed capillary endothelial cells..." Gene expression analysis only predicts functions/roles, and should be stated so.

The danger of overstating correlation as causality is to trivialize future functional studies.

Similarly, as pointed out in the prior review but not addressed, "Aerocytes ... are specialized for gas exchange and leukocyte trafficking, while endothelial gCaps ... promote vascular homeostasis by functioning as progenitor cells, modulating vasomotor tone, and regulating immune activation" needs to be qualified as predicted/implied functions.

Second, the cited alveolar epithelial progenitors (AEP) studies are based on the mouse lung, and are contradicted by single-cell profiling, including this study, showing little evidence for AT2 cell heterogeneity. This discrepancy remains to be resolved and does not corroborate the human data.

This species difference needs to be stated, and the sentence "Both AT2 clusters expressed SFTPC, but one cluster had increased expression of "signaling" genes related to the role of AT2 cells as alveolar epithelial progenitors (AEPs) (hereon called AT2S cells)..." needs to be modified.

Related to this, "we performed in situ hybridization for HHIP, SFTPA1, and immunostaining for SFTPC (Supplemental Figure 5), and confirmed HHIP co-localized with SFTPA1 in a subset of SFTPC+ cells." But Sup. Fig. 5 shows all SFTPA1 cells seem to be SFTPC+ and if so, does not add value to distinguish AT2 cell subsets. Please clarify the relationships among SFTPC, SFTPA1, and HHIP, and label single/double/triple positive cells in the relevant figures.

Line 196: the first SFTPC is misspelled as SFPTC.

Reviewer #2:

Remarks to the Author:

The authors have satisfactorily addressed my previous concerns through extensive new data that support the hypothesis. Congratulations.

Reviewer #3:

Remarks to the Author:

Authors have addressed all comments. I have no further comments.

Dear editor and reviewers,

Thank you again for reading our manuscript. Your time and efforts are appreciated. Below is a point-by-point response to the reviewers.

Reviewer #1 (Remarks to the Author):

The revised manuscript is much improved. Two points need to be clarified in text.

- Thank you for your critical appraisal of the manuscript.

First, the abstract says “Network analyses identified an important role for inflamed capillary endothelial cells...” Gene expression analysis only predicts functions/roles, and should be stated so. The danger of overstating correlation as causality is to trivialize future functional studies.

We agree with this statement by the reviewer. Therefore, we have made the following changes to the manuscript. We changed the wording of the abstract to address the reviewer’s concern.

Chronic obstructive pulmonary disease (COPD) is a leading cause of death worldwide, however our understanding of cell specific mechanisms underlying COPD pathobiology remains incomplete. Here, we analyze single-cell RNA sequencing profiles of explanted lung tissue from subjects with advanced COPD or control lungs, and we validate findings using single-cell RNA sequencing of lungs from mice exposed to 10 months of cigarette smoke, RNA sequencing of isolated human alveolar epithelial cells, functional in vitro models, and in situ hybridization and immunostaining of human lung tissue samples. We identify a subpopulation of alveolar epithelial type II cells with transcriptional evidence for aberrant cellular metabolism and reduced cellular stress tolerance in COPD. Using transcriptomic network analyses, we find evidence suggesting capillary endothelial cells inflammation in COPD, particularly through increased CXCL-motif chemokine signaling. Finally, we detect a high-metallothionein expressing macrophage subpopulation enriched in advanced COPD. Collectively, these findings highlight cell-specific mechanisms involved in the pathobiology of advanced COPD.

We also now begin our discussion of the manuscript’s limitations with highlighting the limitations of transcriptomic analyses.

“There are a few limitations that must be considered while contemplating the findings of this study. First, transcriptional changes are informative but do not always reflect protein concentration or function. While we validated key findings, further studies will be needed to fully elucidate the mechanisms through which the identified transcriptional phenotypes of alveolar cells in COPD contribute to disease pathogenesis.

Similarly, as pointed out in the prior review but not addressed, “Aerocytes ... are specialized for gas exchange and leukocyte trafficking, while endothelial gCaps ... promote vascular homeostasis by functioning as progenitor cells, modulating vasomotor tone, and regulating immune activation” needs to be qualified as predicted/implicit functions.

We apologize for misinterpreting and therefore not addressing the reviewer’s comment during the last review. We have now changed the text of the manuscript regarding this comment. We now write the following:

“We identified all major endothelial populations that have been previously described including arterial, venous, lymphatic, and systemic peri-bronchial endothelial cells, as well as two populations of capillary types. These two capillary cell populations, called aerocytes and general capillaries (gCaps), have the

following ascribed functionality based on recently described transcriptional profiling, lineage tracing, and imaging studies^{37,38} (Figure 4A). Both capillary cell types express PRX (Supplemental Figure 15A), but aerocytes uniquely express CA4 and HPGD and are thought to be specialized for gas exchange and leukocyte trafficking, while endothelial gCaps uniquely express FCN3 and increased levels of NOSTRIN (Supplemental Figure 15B), and are thought to promote vascular homeostasis by functioning as progenitor cells, modulating vasomotor tone, and regulating immune activation.”

Second, the cited alveolar epithelial progenitors (AEP) studies are based on the mouse lung, and are contradicted by single-cell profiling, including this study, showing little evidence for AT2 cell heterogeneity. This discrepancy remains to be resolved and does not corroborate the human data. This species difference needs to be stated, and the sentence “Both AT2 clusters expressed SFTPC, but one cluster had increased expression of “signaling” genes related to the role of AT2 cells as alveolar epithelial progenitors (AEPs) (hereon called AT2S cells)...” needs to be modified.

We agree with the comments of the reviewers. Therefore, we change the text as written below. Notably, we use the same terms as those recently described in the manuscript by Travaglini et al where they identified two populations of AT cells in humans. However, we acknowledge the discrepancies between mouse and human in both the Methods and the Discussion. We have completely revised the paragraph in the results section where we introduce the concept of AT2 cells, and highlight key relevant statements in red.

We identified two clusters of AT2 cells. These two AT2 clusters had similar transcriptional profiles to the two AT2 clusters reported by Travaglini et al in their scRNAseq analysis of normal human lungs, and therefore we used similar terms to annotate these clusters¹⁸. Both AT2 clusters express SFTPC, but one cluster was composed of cells with increased expression of canonical “bulk” AT2 markers such as SFTPA1, SFTPA2, and ETV5 (hereon called AT2_B cells) (Figure 2C, Supplemental Figure 3). AT2_B cells also had increased expression of CA2 and increased expression of WIF1, a negative regulator of WNT signaling. We compared differentially expressed genes (DEGs) between the two AT2 cell populations and found AT2_B cells had DEGs enriched for gene ontology (GO) pathways related to metabolic processes (FDR = 4.18x10⁻³⁹) and biosynthetic processes (FDR = 2.64x10⁻³⁷). The other AT cluster had increased expression of genes related to EGF family receptor signaling (ERBB4), WNT signaling (TNIK, TCF12), and other developmental signaling pathways (FOXP1, STAT3, YAP, TEAD1) (Figure 2C, Supplemental Figure 3). Cells from this cluster (hereon called AT2_S cells) had DEGs enriched for GO pathways related to cell differentiation (FDR = 5.89x10⁻⁶) and cell development (5.92x10⁻⁶). A comparison of the transcriptional profiles of these two AT2 cell populations suggested AT2_S cells may be homologous to AT2 stem and/or alveolar epithelial progenitors (AEPs). However, these similarities are provisional because AEPs are best described in mice and numerous differences between mouse and human AT2 cells have been described¹⁹⁻²¹.

In the Discussion, we have modified the paragraph related to our discussion of AT2 cell types, and highlight key relevant statements in red.

*Our findings highlight an important role for AT2_B cells in COPD pathobiology. AT2_B and AT2_S cells are subtypes of AT2 cells that we identified in our scRNAseq analysis of human lung tissue samples and were recently described by Travaglini et al.¹⁸ AT2_S cells had increased expression of genes related to cellular development and therefore **may** represent AEPs, while AT2_B cells had increased expression of genes related to surfactant production. Interestingly, the largest difference between the two cell populations was increased AT2_B expression of genes implicated in surfactant production and other homeostatic AT2 functions, suggesting that contrasting features between the two AT cell populations are not only related to AT2 progenitor function, and therefore their different properties may extend beyond the eponymous functions of AEPs and non-AEPs. **Additionally, it should be noted that while AEPs are best described in***

mouse, we did not identify two clear AT2 clusters in mice like we identified in humans. There are also substantial differences between mouse and human AT2 cells¹⁹⁻²¹. Therefore, future mechanistic studies will be necessary to elucidate the function of AT2_S and AT2_B cells in the human lung.

Related to this, “we performed *in situ* hybridization for HHIP, SFTPA1, and immunostaining for SFTPC (Supplemental Figure 5), and confirmed HHIP co-localized with SFTPA1 in a subset of SFTPC+ cells.” But Sup. Fig. 5 shows all SFTPA1 cells seem to be SFTPC+ and if so, does not add value to distinguish AT2 cell subsets. Please clarify the relationships among SFTPC, SFTPA1, and HHIP, and label single/double/triple positive cells in the relevant figures.

We found HHIP co-localized with SFTPA1 in many, but not all SFTPC⁺ cells. We have revised the text to reflect the results more accurately and added arrows to clarify the figure. In addition to the figure below, we now write

“We found HHIP expression co-localized with SFTPA1 in a subset of SFTPC⁺ cells, but not all SFTPC⁺ cells express HHIP and SFTPA1.”

Supplemental Figure 5: Co-localization of HHIP and SFTPA1 in AT2 cells. Immunofluorescence staining for pro-surfactant protein C (SFTPC) (purple) *in situ* hybridization for HHIP (red), SFTPA1 (green), and DAPI (blue) in normal human lung tissue samples. Bar = 100 μ m. Original magnification, $\times 20$. Inset shows a SFTPC⁺, HHIP⁺, SFTPA1⁺ cell. Yellow arrows point to examples of SFTPC⁺ HHIP⁻ SFTPA1⁻ cells. Images representative of 5 samples.

Line 196: the first SFTPC is misspelled as SFPTC.

Thank you. We have now corrected this mistake.

Reviewer #2 (Remarks to the Author):

The authors have satisfactorily addressed my previous concerns through extensive new data that support the hypothesis. Congratulations.

Thank you and thank you for taking the time to critically appraise our manuscript.

Reviewer #3 (Remarks to the Author):

Authors have addressed all comments. I have no further comments.

Thank you and thank you for taking the time to critically appraise our manuscript.

Twitter: @maorsauler, @jmcdonou, @KaminskiMed, @ivanorosas, @YaleIMed, @YaleMed @YalePCCSM

Brief Summary of main findings: Single-cell RNA sequencing of human lung tissue identifies transcriptional changes in alveolar niche cells associated with advanced Chronic Obstructive Pulmonary Disease.